# A chemical screen in zebrafish embryonic cells establishes that Akt activation is required for neural crest development

Christie Ciarlo[1,2], Charles K Kaufman[3,4], Beste Kinikoglu[5,6], Jonathan Michael[1], Song Yang[1], Christopher D'Amato[1], Sasja Blokzijl-Franke[7], Jeroen den Hertog[7], Thorsten M Schlaeger[1], Yi Zhou[1], Eric Liao[2,5,6,8], Leonard I Zon[1,2,8]*

[1]Stem Cell Program and Hematology/Oncology, Children's Hospital Boston, Howard Hughes Medical Institute, Boston, United States; [2]Harvard Medical School, Boston, United States; [3]Division of Oncology, Department of Medicine, Washington University School of Medicine, St. Louis, United States; [4]Department of Developmental Biology, Washington University School of Medicine, St. Louis, United States; [5]Center for Regenerative Medicine, Massachusetts General Hospital, Boston, United States; [6]Division of Plastic and Reconstructive Surgery, Massachusetts General Hospital, Boston, United States; [7]Hubrecht Institute, Koninklijke Nederlandse Akademie van Wetenschappen, University Medical Center Utrecht, Utrecht, Netherlands; [8]Harvard Stem Cell Institute, Cambridge, United States

**Abstract** The neural crest is a dynamic progenitor cell population that arises at the border of neural and non-neural ectoderm. The inductive roles of FGF, Wnt, and BMP at the neural plate border are well established, but the signals required for subsequent neural crest development remain poorly characterized. Here, we conducted a screen in primary zebrafish embryo cultures for chemicals that disrupt neural crest development, as read out by *crestin:EGFP* expression. We found that the natural product caffeic acid phenethyl ester (CAPE) disrupts neural crest gene expression, migration, and melanocytic differentiation by reducing Sox10 activity. CAPE inhibits FGF-stimulated PI3K/Akt signaling, and neural crest defects in CAPE-treated embryos are suppressed by constitutively active Akt1. Inhibition of Akt activity by constitutively active PTEN similarly decreases *crestin* expression and Sox10 activity. Our study has identified Akt as a novel intracellular pathway required for neural crest differentiation.

DOI: https://doi.org/10.7554/eLife.29145.001

*For correspondence:
zon@enders.tch.harvard.edu

## Introduction

The neural crest is an embryonic progenitor cell population common to all vertebrates. These cells are highly migratory and give rise to tissues canonically derived from both ectoderm and mesoderm, including bone and cartilage of the head, peripheral neurons, and pigment cells. Neural crest specification begins during gastrulation with the expression of neural plate border specifiers. As development proceeds to the neurula stage, these transcription factors in turn regulate the expression of neural crest specifiers, which mark cells that will migrate and differentiate into neural crest derivatives (*Sauka-Spengler and Bronner-Fraser, 2008*). Not all neural plate border cells are fated to become neural crest, raising the question of which signals regulate the decision between neural crest and other ectodermal derivatives.

In several cases the same signals that regulate neural plate border specification can later regulate neural crest specification. Reiterated Wnt signaling plays a role in neural crest development in chicks, frogs, zebrafish (*García-Castro et al., 2002*; *Monsoro-Burq et al., 2005*; *Lewis et al., 2004*; *Sato et al., 2005*). Wnt acts early in neural crest development to regulate expression of the neural plate border genes *pax3* and *msx1*, as well as later to allow *pax3* and *zic1* to activate transcription of neural crest specifiers including *foxd3* and *slug* (*Lewis et al., 2004*; *Sato et al., 2005*). BMP is also reported to play a reiterated role in neural crest development. In *Xenopus* attenuation of BMP signaling by Hairy2 upregulates neural plate border genes but inhibits neural crest genes (*Nichane et al., 2008*).

While much work has contributed to our knowledge of morphogens required for neural crest induction, less is known about the intracellular signals that are activated in response to these ligands. Fibroblast growth factor (FGF) is reported to play both a cell autonomous and non-cell autonomous role in neural crest induction, either by directly inducing neural crest gene expression or by inducing Wnt8 expression in the paraxial mesoderm (*Hong et al., 2008*; *Yardley and García-Castro, 2012*; *Stuhlmiller and García-Castro, 2012*). FGFs can activate four major intracellular pathways: MAPK, AKT, PLCγ, and STAT (*Turner and Grose, 2010*). Which of these are important during neural crest has not been systematically addressed, though several studies have shown that MAPK signaling acts downstream of FGF in early neural crest induction (*Stuhlmiller and García-Castro, 2012*; *Martínez-Morales et al., 2011*).

Akt, also referred to as protein kinase B, is a critical effector downstream of receptor tyrosine kinases. Classically studied for its oncogenic properties, Akt and its upstream activator PI3-kinase (PI3K) play an important role in cell survival and cell cycle progression. Akt also plays a role in the development of many tissues, canonically acting through negative regulation of FoxO transcription factors (*Accili and Arden, 2004*). The Akt pathway has been particularly well-studied in the context of myogenic differentiation, where it induces myoblast fusion (*Jiang et al., 1998*). Akt also regulates β-catenin, promoting its transcriptional activity by both direct and indirect phosphorylation (*Fang et al., 2007*).

In this study we took advantage of chemical screening in zebrafish to better understand pathways regulating neural crest development. We developed a heterogeneous neural crest cell culture system to screen for chemicals that specifically decrease expression of the neural crest marker *crestin:EGFP*. We found that caffeic acid phenethyl ester (CAPE) inhibits expression of *crestin* by reducing Sox10 activity. CAPE also disrupts neural crest migration and decreases formation of pigmented melanocytes. We found that CAPE inhibits FGF-stimulated PI3K/Akt signaling in vitro, and expression of constitutively active Akt1 suppresses the effects of CAPE on the neural crest in vivo. Reduction of Akt activity by constitutively active PTEN similarly decreases *crestin* expression. We have identified PI3K/Akt as a novel intracellular pathway required for neural crest differentiation through regulation of Sox10 activity.

## Results

### An in vitro screen for chemicals that decrease *crestin:EGFP* expression

To better understand the signals essential for neural crest development, we looked for small molecules that decreased expression of the neural crest reporter *crestin_1 kb:EGFP* (hereafter referred to as *crestin:EGFP*). The 1 kb *crestin* promoter fragment recapitulates endogenous *crestin* mRNA expression, thus marking the neural crest lineage in vivo (*Kaufman et al., 2016*). We developed a neural crest culture protocol to facilitate rapid and automated chemical screening while maintaining this transient cell population in heterogeneous cultures (*Figure 1A,B*) (*Ciarlo and Zon, 2016*). This approach allowed us to distinguish broadly toxic chemicals from those with selective effects on the neural crest. *Crestin:EGFP; ubi:mCherry* transgenic zebrafish embryos were grown to the 5 somite stage (ss), mechanically homogenized, and plated on standard tissue culture-coated plastic in media optimized for neural crest growth and survival, containing fetal bovine serum (FBS), epidermal growth factor (EGF), fibroblast growth factor 2 (FGF2), and insulin (*Kinikoglu et al., 2014*). Under these conditions, *crestin:EGFP*+ cells arose and proliferated, accounting for approximately 20% of total cells after 24 hr in culture (*Figure 1—figure supplement 1A–B*). These cells were highly

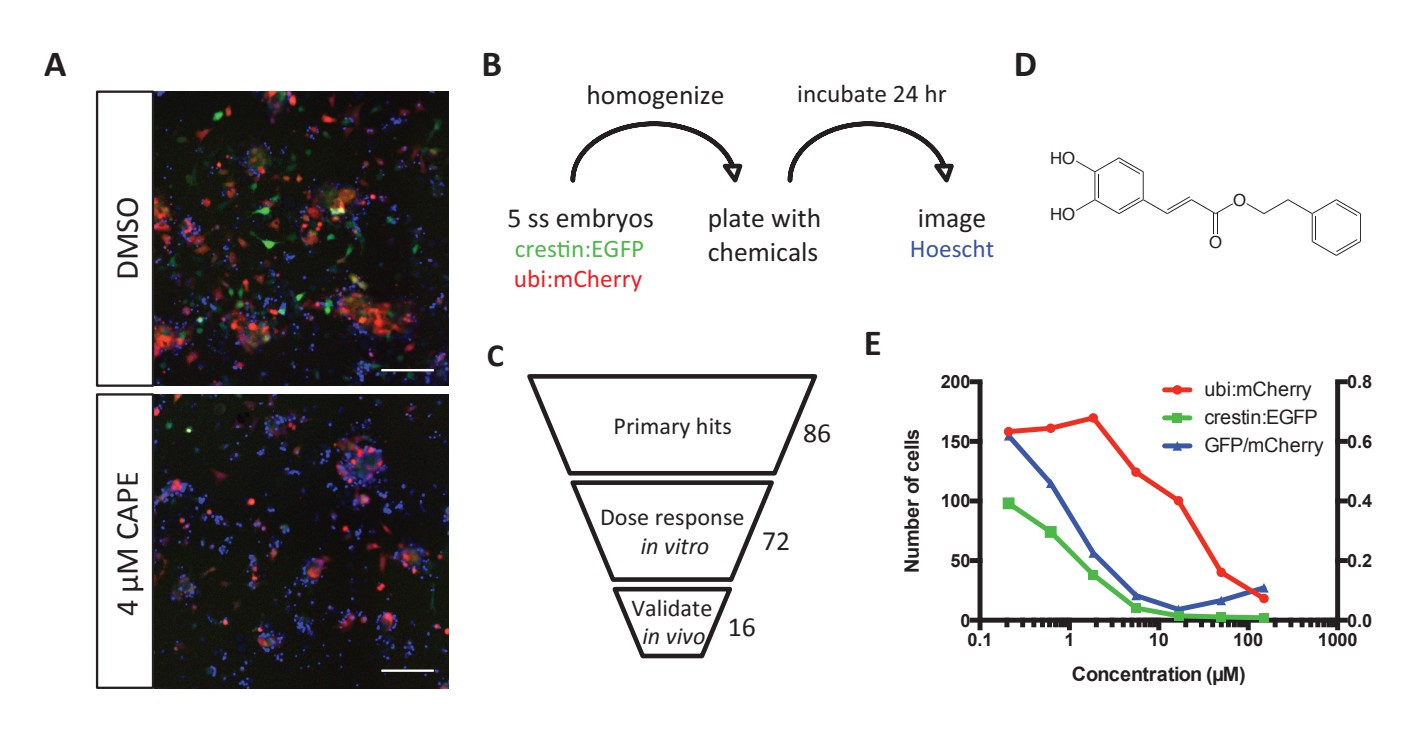

**Figure 1.** Chemical screening in zebrafish embryonic cell cultures identifies inhibitors of neural crest development. (**A**) CAPE decreases *crestin:EGFP* + cells in culture while leaving *ubi:mCherry+* cells unchanged. Scale bar: 100 μm. Characterization of cultured *crestin:EGFP+* cells is shown in *Figure 1—figure supplement 1*. (**B**) Chemical screen design. Whole embryos were mechanically homogenized and plated on standard tissue culture coating in neural crest medium with chemicals. After 24 hr of culture, Hoechst 33342 was added to stain nuclei before imaging. (**C**) Screen hits were validated in the primary screen assay and tested for an effect on *crestin:EGFP* expression in vivo. (**D**) Structure of CAPE. (**E**) CAPE shows a greater than 10-fold selectivity for *crestin:EGFP+* cells versus *ubi:mCherry+* cells in vitro.

DOI: https://doi.org/10.7554/eLife.29145.002

The following source data and figure supplements are available for figure 1:

**Source data 1.** *Ubi:mCherry* and *crestin:EGFP+* cell numbers in CAPE-treated cultures.
DOI: https://doi.org/10.7554/eLife.29145.004

**Figure supplement 1.** Characterization of cultured *crestin:EGFP+* cells.
DOI: https://doi.org/10.7554/eLife.29145.003

**Figure supplement 1—source data 1.** Percentage *crestin:EGFP+* cells in culture.
DOI: https://doi.org/10.7554/eLife.29145.005

**Figure supplement 1—source data 2.** Cell migration speed.
DOI: https://doi.org/10.7554/eLife.29145.006

**Figure supplement 1—source data 3.** Fraction EdU+ cells.
DOI: https://doi.org/10.7554/eLife.29145.007

**Figure supplement 1—source data 4.** qPCR analysis of neural crest gene expression in cultured *crestin:EGFP+* cells.
DOI: https://doi.org/10.7554/eLife.29145.008

proliferative and migratory, expressed key neural crest genes, and could differentiate into pigmented melanocytes in vivo (*Figure 1—figure supplement 1C–F* and *Video 1*).

We used this primary cell culture system to screen 3400 compounds at two concentrations, including bioactives libraries, FDA approved drugs, and Chembridge novel compounds. Chemical hits were identified based on toxicity and specificity cutoffs, taking into account both total cell number as determined by Hoechst staining of nuclei and *ubi:mCherry+* cells representing a random population of cells (*Figure 1A,B*). All hits were verified by eye, resulting in a hit rate of 1.0–1.8%. Of 86 non-redundant, commercially available hit chemicals, 72 reproduced a specific, dose-dependent decrease in number of *crestin:EGFP+* cells in culture (*Figure 1C*, *Table 1*). We next tested the effect of hit chemicals on *crestin:EGFP* transgenic zebrafish embryos treated at 2 ss (10.6 hpf), early in

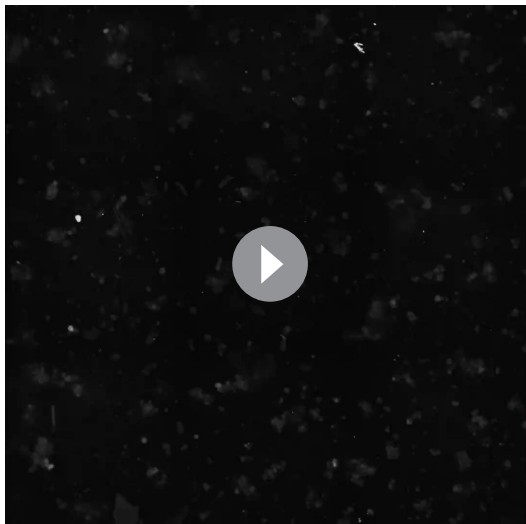

**Video 1.** *Crestin:EGFP* time lapse in heterogeneous neural crest cultures.
DOI: https://doi.org/10.7554/eLife.29145.009

neural crest specification but after the neural plate border is established. Sixteen chemicals decreased *crestin:EGFP* in vivo, including CAPE, a natural product with multiple reported biological activities and targets (*Figure 1D–E*, *Table 1*).

## CAPE disrupts neural crest gene expression in vivo by reducing Sox10 activity

Embryos treated with 10 µM CAPE at 2 ss showed a consistent and dramatic reduction in *crestin:EGFP* expression (*Figure 2A*). We saw a similar effect using the minimal 296 base pair *crestin* promoter (*crestin_296bp:EGFP*) (*Figure 2—figure supplement 1A*). To determine whether CAPE affects *crestin:EGFP* during or after neural crest specification, we conducted time lapse microscopy of *crestin:EGFP* transgenic embryos. In treated embryos, *crestin: EGFP* arose at a severely reduced level, and most *crestin:EGFP+* cells that did emerge quickly disappeared, indicating that they had either died or turned off *crestin:EGFP* expression (*Videos 2* and *3*). Time course experiments confirmed that CAPE acts quickly and early in neural crest development. As determined by whole mount in situ hybridization (ISH), *crestin* expression was decreased after two hours of CAPE treatment, and CAPE had no effect on *crestin* expression in embryos treated at 15 ss or later (*Figure 2—figure supplement 1B–C*) .

We further analyzed the effect of CAPE on neural crest gene expression by ISH. Embryos were treated with CAPE at 2 ss and fixed at 15–17 ss for analysis. We found a decrease in expression of *crestin*, *sox10*, *pax7a*, *dlx2a*, and the Ap-2α target gene *inka1a* (*Figure 2B,C*). These genes are expressed in both premigratory and migratory neural crest. We observed little to no change in the levels of *nr2f2*, *msxb*, *pax3a*, *ets1*, *ap2a*, *ap2c*, *snai1b*, dlx5a, and *foxd3* (*Figure 2—figure supplement 2A,B*). In addition to being expressed in the premigratory and migratory neural crest, a subset of these genes are expressed earlier in development at the neural plate border (*msxb*, *pax3a*, *dlx5a*). Our data indicate that neural crest-like cells are present in CAPE-treated embryos but exhibit abnormal gene expression, particularly for genes expressed after the neural plate border is established. Later in development, at 24 hpf, CAPE treatment caused a dramatic decrease in pigment cell-specific gene expression, including *mitfa* and its target gene *dct*, which mark melanoblasts, and *fms*, which marks xanthophore precursors (*Figure 2D*).

To confirm that changes in neural crest gene expression were not the result of reduced cell number, we evaluated proliferation and cell death in CAPE-treated embryos. After 6 hr of treatment, a time point at which neural crest gene expression is decreased by CAPE, no change in proliferation was observed in *sox10:GFP+* cells, as determined by phospho-histone H3 staining (*Figure 2—figure supplement 3A,B*). To evaluate cell death, we used whole mount TUNEL staining combined with *crestin* ISH. We found a slightly increased number of overall TUNEL positive cells in embryos treated with 10 µM CAPE for 6 hr, but these cells were not present in the region of *crestin* expression, and the magnitude of increased cell death does not explain the differences in *crestin* expression (*Figure 2—figure supplement 3C,D*). After 19 hr of treatment, 10 µM CAPE increased cell death broadly throughout the embryo, most dramatically in the eye and tail, while 5 µM CAPE had no effect on cell death, even as late as 48 hpf (*Figure 2—figure supplement 3E,F*). These data indicate that increased cell death or reduced proliferation do not contribute to the early effects of CAPE on neural crest gene expression. The observed deficiency in pigment cell gene expression is also not explained by increased cell death, since this deficiency was observed with 5 µM CAPE treatment.

To confirm CAPE's effect on neural crest gene expression and identify potential mechanisms of action, we conducted RNA-seq on sorted *sox10:Kaede+* cells from control and CAPE-treated embryos at 17 ss. This data confirmed that neural crest gene expression is decreased in cells from

**Table 1.** In vitro validated screen hits.
Hits that also validated in vivo are bold.

| Compound | Library | Target/category |
|---|---|---|
| **CAPE** | ICCB | NFkB/multiple |
| raloxifene | LOPAC | Estrogen receptor |
| mianserin | LOPAC | 5-HT receptor antagonist |
| GANT61 | LOPAC | Hedgehog |
| MnTBAP | ICCB | SOD mimetic |
| loperamide | FDA approved | Mu opiod receptor agonist |
| latanoprost | FDA approved | prostaglandin F2a analogue |
| tetraethylthiuram disulfide | LOPAC | alcohol dehydrogenase |
| dopamine | LOPAC | dopamine receptor |
| mycophenolate mofetil | LOPAC | IMPH |
| genistein | LOPAC | kinase inhibitor |
| albendazole | FDA approved | antihelminthic |
| JFD00244 | LOPAC | sirt2 inhibitor |
| perphenazine | FDA approved | 5-HT receptor |
| 5-NOT | FDA approved | 5-HT agonist |
| SKF95282 | LOPAC | histamine H2 receptor antagonist |
| bicalutamide | FDA approved | anti-androgen |
| **capsazepine** | LOPAC | sodium channels |
| triflupromazine | LOPAC | monoamine transporters |
| flubendazole | FDA approved | antihelminthic |
| GDC-0941 | LOPAC | PI3K |
| imatinib | FDA approved | RTK inhibitor |
| indatraline | LOPAC | dopamine uptake inhibitor |
| **MBCQ** | ICCB | PDE5 |
| **MDL-28170** | ICCB | calpain inhibitor |
| NS8593 | LOPAC | potassium channels |
| **NU6027** | LOPAC | ATR/CDK2 inhibitor |
| PD180970 | LOPAC | RTK inhibitor |
| PD173074 | LOPAC | src inhibitor |
| PI-103 | LOPAC | PI3K |
| rapamycin | LOPAC | mTOR |
| SB242084 | LOPAC | 5-HT receptor antagonist |
| **triptolide** | FDA approved | RNA pol II |
| tyrphostin AG698 | LOPAC | tyrosine kinase inhibitor |
| wiskostatin | LOPAC | actin |
| PAC-1 | LOPAC | proapoptotic zinc chelator |
| PD407824 | LOPAC | chk1 inhibitor |

*Table 1 continued on next page*

*Table 1 continued*

| Compound | Library | Target/category |
| --- | --- | --- |
| PD173952 | LOPAC | src inhibitor |
| sanguinarine | LOPAC | Na/K ATPase |
| tyrphostin AG835 | LOPAC | EGFR |
| (-)-alpha-methylnorepinephrine | LOPAC | sympathomimetic |
| chloroquine | LOPAC | antimalarial |
| M-344 | LOPAC | HDAC inhibitor |
| olmesartan medoxomil | FDA approved | angiotensin II receptor antagonist |
| **1,10-phenanthroline** | LOPAC | chelator, MMP |
| **2,3-dimethoxy-1,4-naphthoquinone** | LOPAC | ROS |
| amiloride | ICCB | calcium channels |
| fluvastatin | FDA approved | HMG co-A reductase |
| CHM-1 | LOPAC | antimitotic |
| SAHA | LOPAC | HDAC inhibitor |
| **nimesulide** | LOPAC | COX-2 |
| mibefradil | LOPAC | calcium channels |
| KB-R7493 | LOPAC | sodium calcium exchanger |
| LY165163 | LOPAC | 5-HT receptor antagonist |
| dequalinium | LOPAC | potassium channels |
| AM92016 | ICCB | potassium channels |
| 2-[4-(1,3-benzodioxol-5-yl)—1H-pyrazol-1-yl]-N-(2-ethyl-2H-1,2,3-triazol-4-yl)acetamide | Chembridge | predicted adenosine kinase |
| N-(2-ethyl-2H-1,2,3-triazol-4-yl)—2-{4-[3-(1H-pyrazol-1-yl)phenyl]—1H-pyrazol-1-yl}acetamide | Chembridge | predicted adenosine kinase |
| 2,2,6,6-tetramethyl-N-(1-methyl-3-phenylpropyl)—4-piperidinamine | Chembridge | predicted vitamin D receptor |
| N-[(5-chloro-1H-indol-2-yl)methyl]—2-(3-hydroxyphenyl)acetamide | Chembridge | predicted TK(FLT3) |
| 5-(1H-indol-2-ylcarbonyl)—4,5,6,7-tetrahydrothieno[3,2 c]pyridine | Chembridge | predicted TK(FLT3) |
| 1-acetyl-4-{4-[1-(2-fluorophenyl)—1H-pyrazol-4-yl]pyrimidin-2-yl}—1,4-diazepane | Chembridge | predicted JNK |
| **4-(4-butyl-1H-1,2,3-triazol-1-yl)—1-{[(1S*,4S*)—3,3-dimethyl-2-methylenebicyclo[2.2.1]hept-1-yl] carbonyl}piperi** | Chembridge | predicted liver X receptor |
| **1-(3-methylbenzyl)—4-thieno[2,3-d]pyrimidin-4-yl-2-piperazinone** | Chembridge | predicted TK(EGFR, PDGFR, CSFR1); PKC; PKA |
| **1-(2-methoxyphenyl)—2,2-dimethyl-4-(4-methylpentanoyl)piperazine** | Chembridge | predicted androgen receptor |
| **5,6-dimethyl-2-[4-({methyl[(2-methylpyridin-4-yl)methyl]amino}methyl)phenyl]pyrimidin-4(3 hr)-one** | Chembridge | predicted estrogen receptor |
| **N-[1-(1,5-dimethyl-1H-pyrazol-4-yl)ethyl]thieno[2,3-d]pyrimidin-4-amine** | Chembridge | predicted EGFR |
| N-(1-cyclohexyl-1H-pyrazol-5-yl)—2-[3-(2-thienyl)—1H-pyrazol-1-yl]acetamide | Chembridge | predicted VEGFR2, EGF/KDR |
| **2-[1-(3-isobutyl-1,2,4-oxadiazol-5-yl)—2-methylbutyl]—1-isoindolinone** | Chembridge | predicted RAR(gamma) |
| 1-propyl-N-{1-[4-(1H-pyrazol-1-yl)phenyl]piperidin-4-yl}piperidin-4-amine | Chembridge | predicted estrogen receptor |
| **5-[5-methyl-4-(1-methyl-1H-pyrazol-4-yl)pyrimidin-2-yl]—4,5,6,7-tetrahydrothieno[3,2 c]pyridine** | Chembridge | predicted TK(VEGFR, KDR, FLK1) |
| 2-[5-(2,6-dimethylphenyl)—1H-indazol-1-yl]-N-(1,3-dimethyl-1H-pyrazol-5-yl)acetamide | Chembridge | predicted TK(PDGFR, EGFR, FGFR) |

DOI: https://doi.org/10.7554/eLife.29145.010

CAPE-treated embryos, though to a lesser extent than observed in ISH experiments (*Figure 2E*). Some genes including *pax3a* and *nr2f2* appeared unchanged by in situ but were downregulated in RNA-seq data (*Figure 2—figure supplement 2C*). *Sox10:Kaede* is more highly expressed in the cranial than trunk neural crest and is also highly expressed in the otic vesicle, potentially explaining the discrepancies between RNA-seq and ISH results. Consistent with ISH at 24 hpf, CAPE decreased

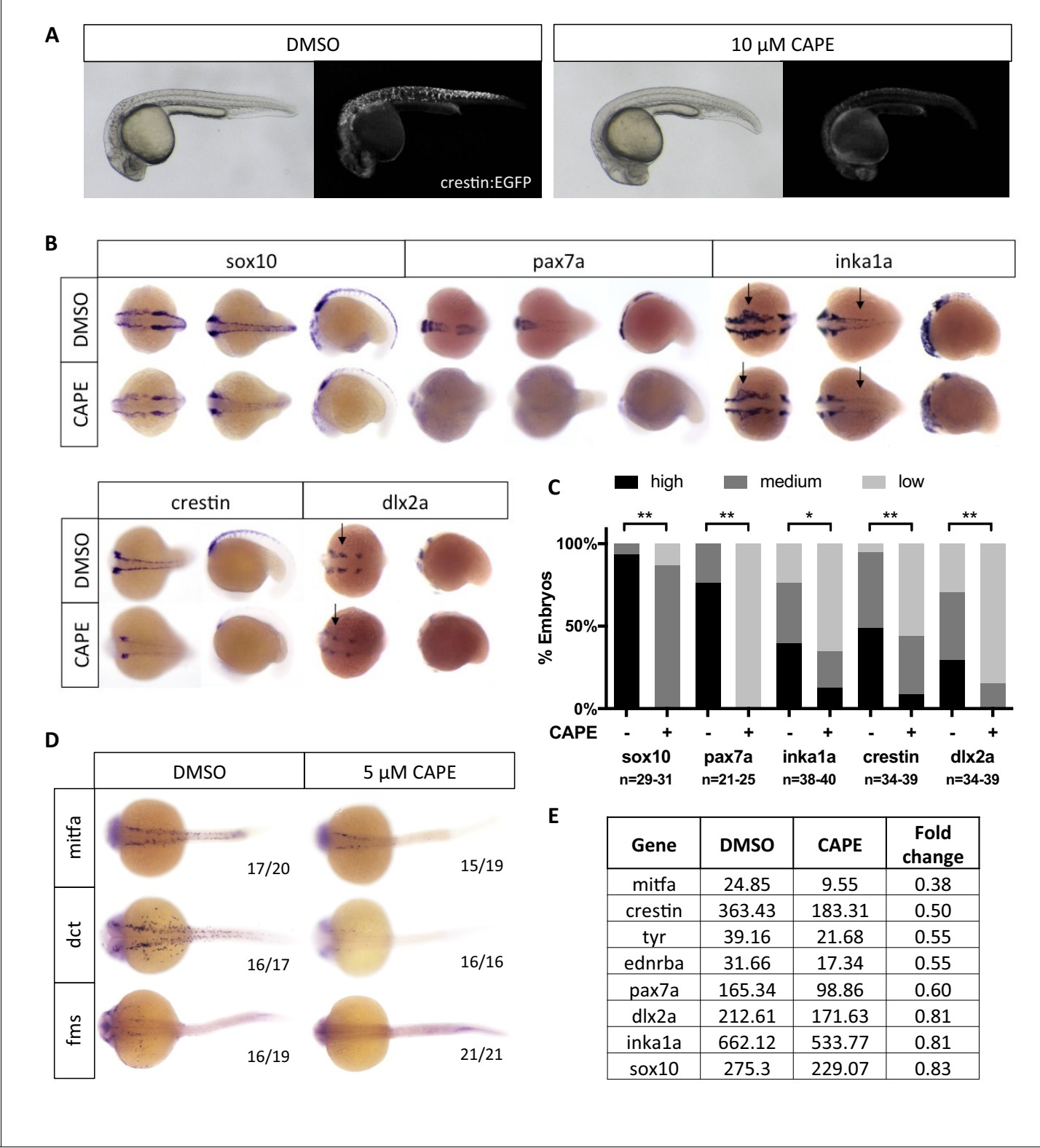

**Figure 2.** CAPE decreases neural crest gene expression. (**A**) CAPE dramatically reduces *crestin:EGFP* expression at 26 hpf. *Figure 2—figure supplement 1* shows the response of a smaller (396 bp) *crestin* promoter fragment to CAPE and the timing of *crestin* response to CAPE. (**B**) CAPE decreases neural crest gene expression as determined by whole mount in situ hybridization (ISH) at 15–17 ss. Expression of some genes is decreased ubiquitously while others are only altered in specific anatomic locations (arrows). Images are representative of at least two independent experiments. (**C**) Scoring of ISH in (**B**). (**D**) CAPE reduces expression of pigment cell genes at 24 hpf. (**E**) FPKM values from RNA-sequencing of *sox10:Kaede+* cells

*Figure 2 continued on next page*

*Figure 2 continued*

confirmed decreases in neural crest genes, though to a lesser extent than by ISH. Cells were sorted from control or CAPE-treated embryos at 17 ss. *Figure 2—figure supplement 2* shows neural crest genes with no significant change by ISH. *Figure 2—figure supplement 3* shows that a change in cell number does not account for neural crest gene expression changes. *Figure 2—figure supplement 4* shows other gene expression changes in neural crest cells. *p<0.001, **p<0.0001, chi-square test.

DOI: https://doi.org/10.7554/eLife.29145.011

The following source data and figure supplements are available for figure 2:

**Source data 1.** Expression of neural crest genes by ISH in CAPE-treated embryos.
DOI: https://doi.org/10.7554/eLife.29145.016

**Figure supplement 1.** *Crestin_296bp:EGFP* expression in CAPE-treated embryos and time course of CAPE treatment.
DOI: https://doi.org/10.7554/eLife.29145.012

**Figure supplement 1—source data 1.** Scoring of *crestin_296bp:EGFP* expression in CAPE-treated embryos.
DOI: https://doi.org/10.7554/eLife.29145.017

**Figure supplement 2.** Neural crest genes not significantly affected by CAPE treatment as determined by ISH.
DOI: https://doi.org/10.7554/eLife.29145.013

**Figure supplement 2—source data 1.** Scoring of neural crest gene expression by ISH in CAPE-treated embryos.
DOI: https://doi.org/10.7554/eLife.29145.018

**Figure supplement 3.** Changes in cell number do not explain reduced *crestin* expression in CAPE-treated embryos.
DOI: https://doi.org/10.7554/eLife.29145.014

**Figure supplement 3—source data 1.** PH3+ cells in *sox10:GFP*+ region of CAPE-treated embryos.
DOI: https://doi.org/10.7554/eLife.29145.019

**Figure supplement 3—source data 2.** TUNEL+ cells in neural crest region of CAPE-treated embryos.
DOI: https://doi.org/10.7554/eLife.29145.020

**Figure supplement 4.** Analysis of gene expression changes in *sox10:Kaede*+ cells by RNA-seq, comparing DMSO- to CAPE-treated embryos.
DOI: https://doi.org/10.7554/eLife.29145.015

expression of melanocyte-specific genes including *mitfa*, *ednrba*, and *tyr* (*Figure 2E*). Ingenuity Pathway Analysis (Qiagen) pointed to an increase in inflammatory signaling and changes in levels of morphogens such as Wnts, BMPs, and FGFs with CAPE treatment (*Figure 2—figure supplement 4*). These results suggest that CAPE's mechanism of action involves cell-cell signaling at the level of secreted ligand expression.

To better understand the mechanism by which CAPE leads to changes in gene expression, we conducted ATAC-seq (Assay for Transposase-Accessible Chromatin) in *sox10:Kaede*+ cells isolated from control and CAPE-treated embryos (*Buenrostro et al., 2013*). We observed a reduction in chromatin accessibility specifically at the *mitfa* promoter with CAPE treatment (*Figure 3A*). Changes in *mitfa* expression and chromatin accessibility could be downstream of Sox10, as Sox10 is known to bind to the promoter of *mitfa* and regulate its expression (*Elworthy et al., 2003*). Indeed, ChIP-seq in a zebrafish melanoma cell line (zcrest 1) showed strong Sox10 binding at the *mitfa* promoter

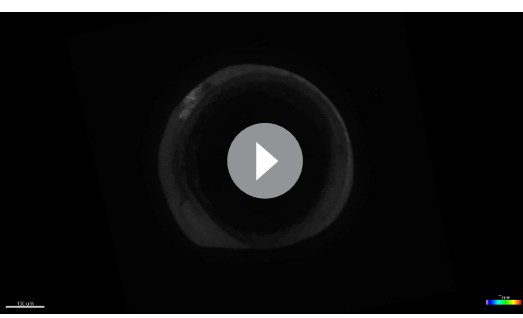

**Video 2.** C*restin:EGFP* expression in DMSO-treated zebrafish embryos. Embryos were treated at 2 ss and mounted for imaging at 10 ss. Embryos were imaged for 16.25 hr, and images were collected every 9 min.
DOI: https://doi.org/10.7554/eLife.29145.021

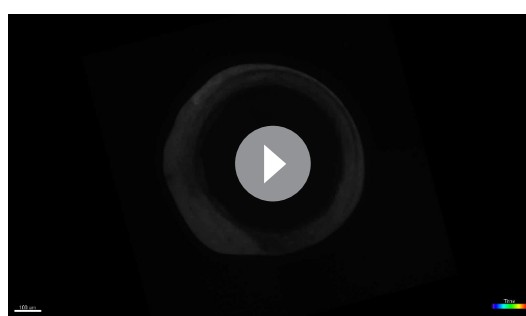

**Video 3.** C*restin:EGFP* expression in CAPE-treated (10 µM) zebrafish embryos. Embryos were treated at 2 ss and mounted for imaging at 10 ss. Embryos were imaged for 16.25 hr, and images were collected every 9 min.
DOI: https://doi.org/10.7554/eLife.29145.022

(*Figure 3A*) (*Kaufman et al., 2016*). Other Sox10-bound genes including *crestin* did not show a reduction in chromatin accessibility even though their expression was decreased (*Figure 3B*). Therefore chromatin closing is likely not the primary mechanism by which CAPE decreases gene expression, but we used the ATAC-seq dataset to identify transcription factors of potential importance to CAPE's mechanism.

To identify transcription factors that could mediate CAPE's effects on gene expression in a genome-wide unbiased manner, we used Hypergeometric Optimization of Motif EnRichment (HOMER) (*Heinz et al., 2010*). Of 81,140 total ATAC peaks, we identified 585 peaks (0.7%) unique to the DMSO control sample as compared to the CAPE-treated sample. We used these peak sequences as input to HOMER, with all peak sequences in the CAPE-treated sample as background. The most enriched motifs were those of Sox transcription factors, including Sox10, and MITF (*Figure 3C*). To determine whether CAPE affects Sox10 transcriptional activity, we overexpressed *sox10* in zebrafish embryos by RNA microinjection. While *sox10* injection increased *crestin:EGFP* expression in untreated embryos, it had no effect on *crestin:EGFP* expression in CAPE-treated embryos, indicating that CAPE reduces Sox10 activity (*Figure 3D–E*). Reduced expression of *crestin:EGFP* in CAPE-treated embryos was not due to a reduction in *sox10*-expressing cells, as the percentage of *sox10:Kaede*+ cells was identical in treated and control embryos (*Figure 3—figure supplement 1C*). In contrast to *sox10*, injection of *tfap2c* RNA increased *crestin:EGFP* in both control and CAPE-treated embryos, though its effect was subtle in both cases (*Figure 3—figure supplement 1A,B*). These data support a role for Sox10 in mediating the transcriptional effects of CAPE.

## CAPE inhibits neural crest migration and pigment cell differentiation in vivo

We found that neural crest migration is also disrupted by CAPE. While *crestin:EGFP* is dramatically reduced upon CAPE treatment, *sox10:GFP* intensity is indistinguishable between control and treated embryos. This may be due to the strength of the *sox10* promoter or persistence of stable GFP. *Sox10:GFP* transgenic embryos allowed us to follow neural crest cells after CAPE treatment. These cells failed to reach the ventral half of the trunk in CAPE-treated embryos (*Figure 4A*, *Videos 4* and *5*). The cells remained rounded instead of extending projections and elongating ventrally, as was observed in control embryos (*Videos 4* and *5*). At 2 days post fertilization (dpf), pigmentation in CAPE-treated embryos was dramatically reduced, indicating that CAPE interferes with the production of differentiated melanocytes (*Figure 4B,C*). A migration defect was also apparent in the increased fraction of dorsal melanocytes at 2 dpf (*Figure 4D*). We found that CAPE's effect on neural crest migration co-occurred with decreased *mitfa:GFP* expression. A melanocyte migration defect was still evident when embryos were treated with CAPE at 15 ss (*Figure 4D*). Therefore the melanocyte migration defect induced by CAPE may be downstream of decreased *mitfa* expression.

We further evaluated the effect of CAPE on xanthophores and iridophores at 3 dpf. We found that CAPE reduces iridophore number, size, and pigmentation, though this effect was not as dramatic as the effect of CAPE on melanocytes (*Figure 4—figure supplement 1A,B*). As for melanocytes, we saw an increase in the fraction of dorsal iridophores, but this effect was again less pronounced than the melanocyte position defect (*Figure 2D*, *Figure 4—figure supplement 1C*). Though we saw reduced *fms* expression at 24 hpf, we found that by 3 dpf, xanthophores had recovered. CAPE-treated embryos in fact showed more intense yellow pigmentation that control embryos, particularly in the head (*Figure 4—figure supplement 1A*). These data indicate that CAPE treatment has selective effects on different pigment cell lineages.

Since Sox10 plays an important role in otic placode development, we evaluated the effect of CAPE on the otic vesicle at 24 and 48 hpf (*Dutton et al., 2009*). We found that CAPE causes a subtle defect in otic vesicle shape at 24 hpf. At 48 hpf, CAPE-treated embryos lack semicircular canal projections and have more closely spaced otoliths (*Figure 4—figure supplement 2*). These defects are consistent with a reduction in Sox10 activity, since they have been observed in *sox10* mutants (*Dutton et al., 2009*).

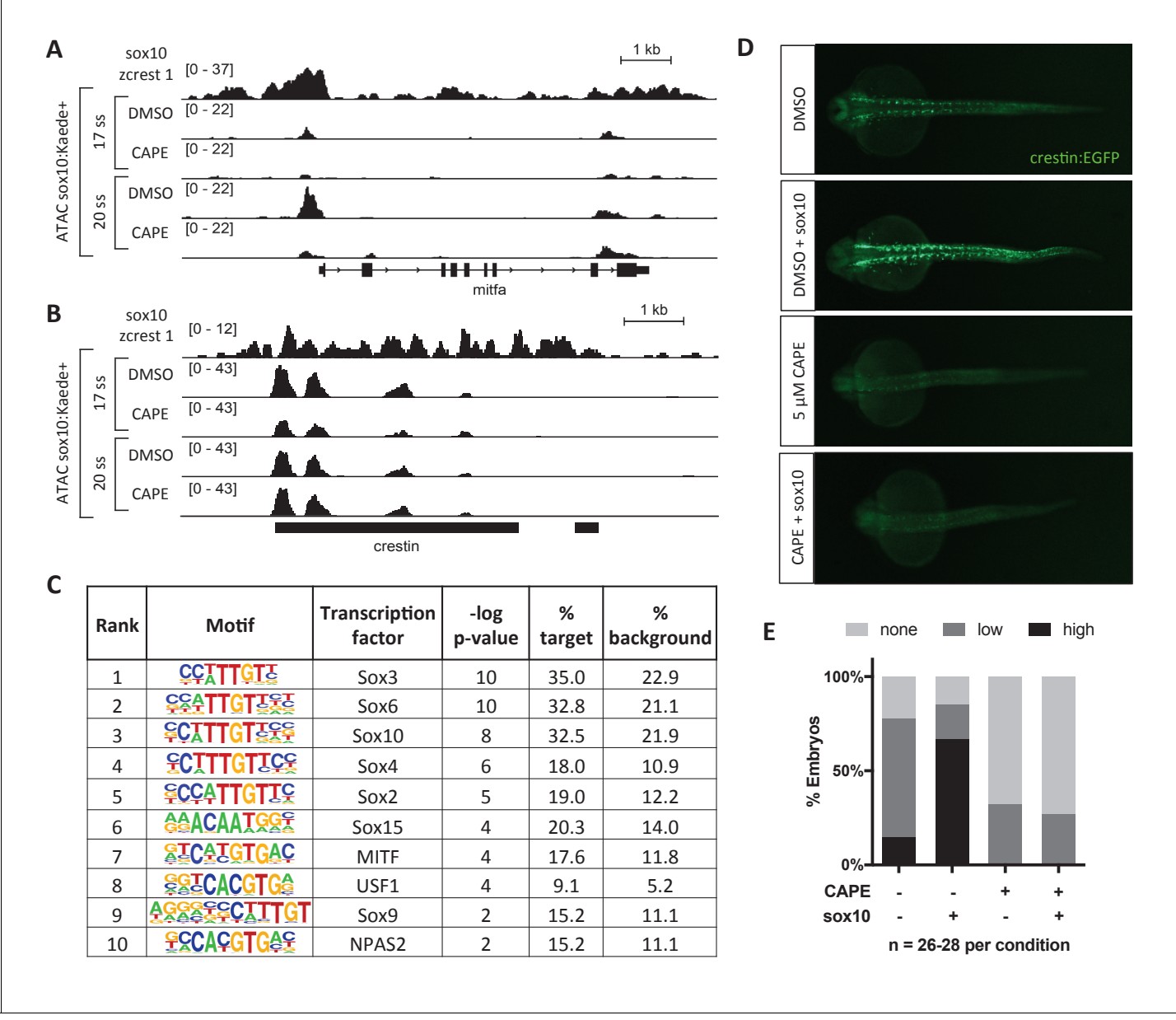

**Figure 3.** CAPE reduces Sox10 activity. (**A**) ATAC-seq was conducted on *sox10:Kaede*+ cells from DMSO- or CAPE-treated embryos at two stages. CAPE reduces chromatin accessibility at the *mitfa* promoter in *sox10:Kaede*+ cells, and Sox10 binds the *mitfa* promoter in a zebrafish tumor cell line. (**B**) *Crestin* binds Sox10 but does not show a change in chromatin accessibility with CAPE treatment. Bar indicates region of *crestin* sequence similarity (chr4:41,270,000). (**C**) HOMER analysis of 20 ss ATAC-seq peaks revealed an enrichment for Sox and MITF motifs when comparing unique peaks in DMSO-treated embryos (% target) to all peaks in CAPE-treated embryos (% background). (**D**) CAPE (5 μM) prevents *sox10* RNA (30 pg) from increasing *crestin:EGFP* expression. (**E**) Quantification of experiment shown in (**D**). Sum of three clutches from two independent experiments is shown. *Figure 3—figure supplement 1* shows that *tfap2c* RNA increases *crestin:EGFP* expression in both DMSO- and CAPE-treated embryos, and that the number of *sox10:Kaede*+ cells does not change with CAPE treatment.
DOI: https://doi.org/10.7554/eLife.29145.023

The following source data and figure supplements are available for figure 3:

**Source data 1.** Scoring of *crestin:EGFP* expression in *sox10*-injected and CAPE-treated embryos.
DOI: https://doi.org/10.7554/eLife.29145.025

**Figure supplement 1.** *Tfap2c* RNA injection (120 pg) increases *crestin:EGFP* expression in both control and CAPE-treated embryos.
DOI: https://doi.org/10.7554/eLife.29145.024

**Figure supplement 1—source data 1.** Scoring of *crestin:EGFP* expression in *tfap2c*-injected and CAPE-treated embryos.
DOI: https://doi.org/10.7554/eLife.29145.026

*Figure 3 continued on next page*

*Figure 3 continued*

**Figure supplement 1—source data 2.** Percentage *sox10:Kaede*+ live cells in CAPE-treated embryos.
DOI: https://doi.org/10.7554/eLife.29145.027

## Inhibition of Akt signaling contributes to CAPE-induced defects in neural crest development

CAPE has previously been reported to inhibit Akt signaling in melanoma cell lines by inhibition of PI3K activity (*Pramanik et al., 2013*). In addition to CAPE, we identified the PI3K inhibitors PI-103 and GDC-0941 as strong hits in our screen and confirmed that they specifically reduce *crestin:EGFP* + cell number in a dose-responsive manner (*Figure 5—figure supplement 1A*). To determine whether inhibition of Akt signaling might contribute to the effects of CAPE on the zebrafish neural crest, we overexpressed constitutively active, membrane-targeted human Akt1 (myr-Akt1) in zebrafish embryos by RNA microinjection. Injection of myr-Akt1 increased Akt phosphorylation independent of CAPE treatment and rescued *crestin* expression in CAPE-treated embryos (*Figure 5A–C*). Myr-Akt1 injection also partially rescued both melanocyte number and melanocyte migration at 2 dpf (*Figure 5D–F*). Defects such as pericardial edema and curved tail showed a slight but consistent decrease with myr-Akt1 injection (*Figure 5—figure supplement 1B*). Overall these data indicate that inhibition of Akt signaling contributes to CAPE's effects on both gene expression and migration of neural crest cells.

To more precisely elucidate the mechanism of Akt inhibition by CAPE, we returned to the in vitro culture system we used for chemical screening. This system provides an opportunity to control chemical cues received by prospective neural crest cells. In addition to FBS, which contains unknown factors required for neural crest induction in vitro, two growth factors in the culture medium promoted neural crest induction: FGF2 and insulin (*Figure 6A,B*). We studied two pathways known to be activated downstream of these factors: PI3K/Akt and Mek/Erk. We found that while insulin primarily activated Akt, FGF primarily activated Erk (*Figure 6C,D*).

Surprisingly, we found that CAPE inhibited Akt phosphorylation only in the presence of FGF, even though FGF itself did not promote Akt activation (*Figure 6C*, *Figure 6—figure supplement 1A*). This was not true for the PI3K inhibitor PI-103 that prevented Akt phosphorylation regardless of FGF stimulation (*Figure 6—figure supplement 1C*). We hypothesized that the apparent lack of Akt stimulation by FGF could be explained by Akt inhibition by another pathway. Addition of a Mek inhibitor to the cultures resulted in a dramatic increase in p-Akt, leading us to postulate that Mek/Erk negatively regulates PI3K/Akt (*Figure 6—figure supplement 1A*, last two lanes). In the context of Mek inhibition, we saw that FGF activated Akt to a similar extent as insulin (*Figure 6—figure supplement 1A*, first three lanes). FGF treatment enhanced the ability of Mek inhibition to stimulate p-Akt (*Figure 6—figure supplement 1A*, compare lanes 4 and 6). Mek inhibition also blunted the ability of CAPE to reduce p-Akt (*Figure 6—figure supplement 1B*, compare lanes 2 and 6). Finally, the induction of p-Akt by Mek inhibition was PI3K-dependent, since it was blocked by a PI3K inhibitor (*Figure 6—figure supplement 1B*, lanes 1–4). These observations led us to formulate a model in which FGF modulates Akt activation through a mechanism distinct from insulin-stimulated Akt activation and sensitive to CAPE activity (*Figure 6—figure supplement 1D*). Since the phosphorylation of membrane-targeted Akt1 (myr-Akt1) is not affected by CAPE, CAPE acts upstream of Akt membrane recruitment by PIP3 (*Figure 5A*). CAPE could function through inhibition of PI3K or activation of Pten, a negative regulator of PIP3. We ruled out the latter hypothesis, since CAPE had the same effect on *crestin* expression in wild type or *ptena-/-;ptenb-/-* embryos (*Figure 6—figure supplement 1E*).

While FGF2 and insulin were used to induce neural crest in culture, we asked what growth factors regulate neural crest development in vivo. Many kinase inhibitors scored as hits in our initial screen, including two chemicals from the Chembridge KINAcore library that were generated as structural analogs to receptor tyrosine kinase (RTK) inhibitors. These two chemicals also decreased *crestin* expression in vivo, while EGFR, FGFR, insulin-like growth factor receptor, and insulin receptor inhibitors had no effect when added at 2 ss (*Figure 6—figure supplement 2*). Importantly we tested SU5402, an FGFR inhibitor with demonstrated activity in zebrafish embryos (*Molina et al., 2007*). This data suggests that a growth factor other than FGF activates Akt in neural crest cells in vivo.

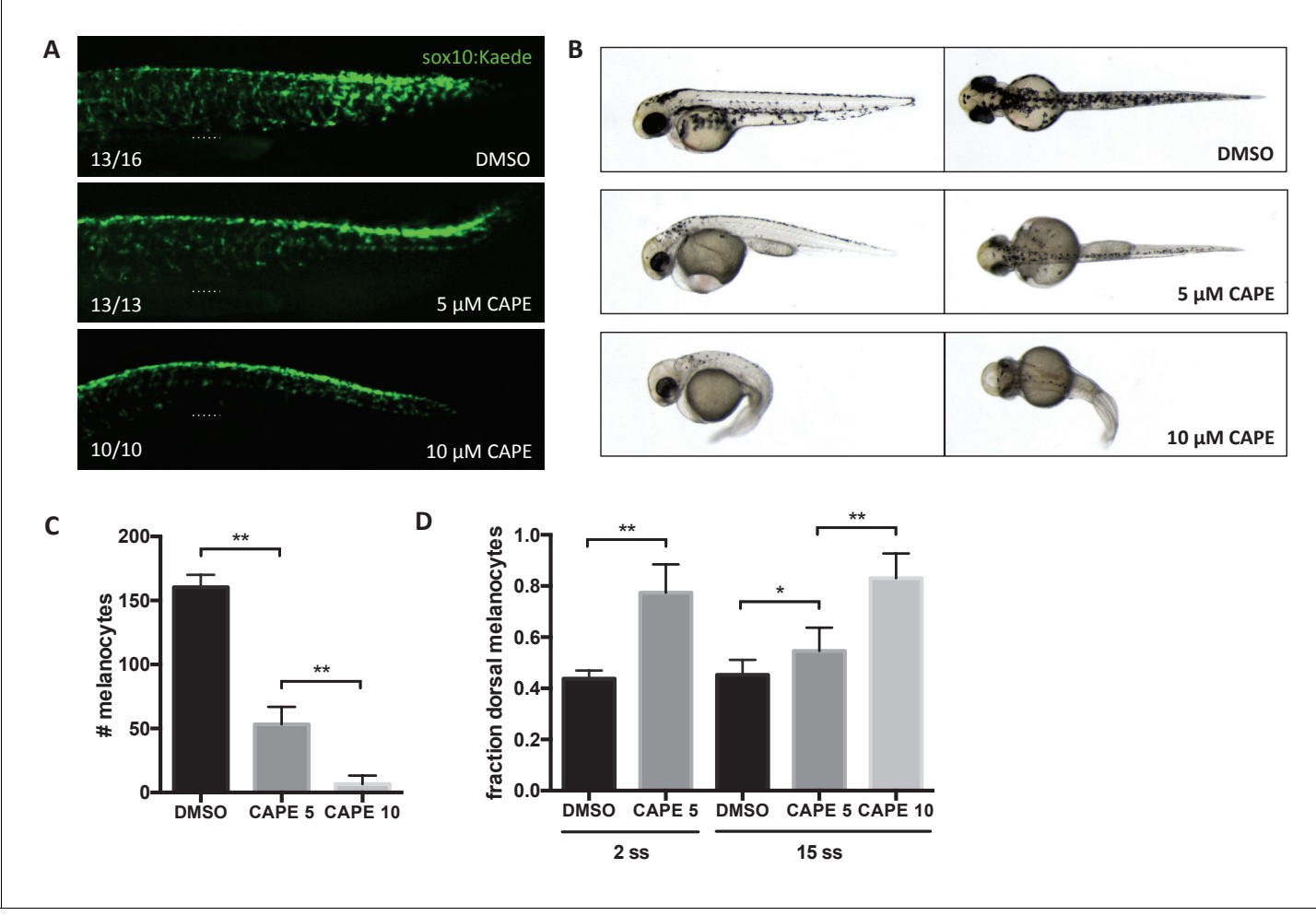

**Figure 4.** CAPE inhibits neural crest migration and pigment cell differentiation. Embryos were treated at 2 ss unless otherwise indicated. (**A**) *Sox10: Kaede+* cells in the trunk of zebrafish embryos are more dorsally located at 24 hpf. Dotted line indicates top of yolk sac extension. CAPE-treated embryos were allowed to develop for 4 hr longer than control embryos for stage matching. Numbers indicate fraction of embryos for which images are representative. Some DMSO control embryos showed a pattern similar to 5 µM CAPE. Similar results were observed in three independent experiments. (**B**) Morphology and pigmentation of CAPE-treated embryos at 2 dpf. Treated embryos showed reduced pigmentation and defects such as pericardial edema and a curved tail. (**C**) Melanocyte counts corresponding to (**B**). Trunk melanocytes were counted from the yolk sac extension to the end of the tail. Error bars represent standard deviation of 10 embryos from two independent experiments. (**D**) CAPE increases the fraction of dorsal melanocytes at 2 dpf. Melanocytes were counted as in (**C**). Embryonic stage at drug treatment is indicated. *p<0.05, **p<0.0005, unpaired t-test. *Figure 4—figure supplement 1* shows the effect of CAPE on xanthophores and iridophores at 3 dpf. *Figure 4—figure supplement 2* shows the effect of CAPE on otic vesicle development.

DOI: https://doi.org/10.7554/eLife.29145.028

The following source data and figure supplements are available for figure 4:

**Source data 1.** Melanocyte numbers in CAPE-treated embryos.
DOI: https://doi.org/10.7554/eLife.29145.031
**Source data 2.** Fraction dorsal melanocytes in CAPE-treated embryos.
DOI: https://doi.org/10.7554/eLife.29145.032
**Figure supplement 1.** CAPE disrupts iridophore development less dramatically than melanocyte development.
DOI: https://doi.org/10.7554/eLife.29145.029
**Figure supplement 1—source data 1.** Number of iridophores in CAPE-treated embryos.
DOI: https://doi.org/10.7554/eLife.29145.033
**Figure supplement 1—source data 2.** Fraction dorsal iridophores in CAPE-treated embryos.
DOI: https://doi.org/10.7554/eLife.29145.034
**Figure supplement 2.** CAPE disrupts ear development.
DOI: https://doi.org/10.7554/eLife.29145.030

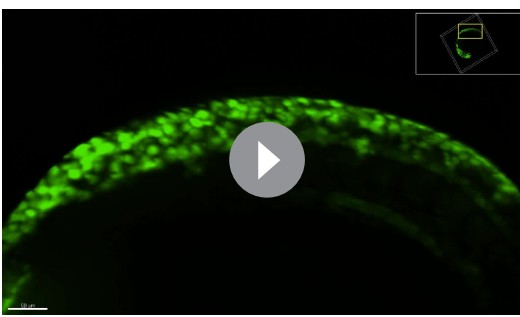 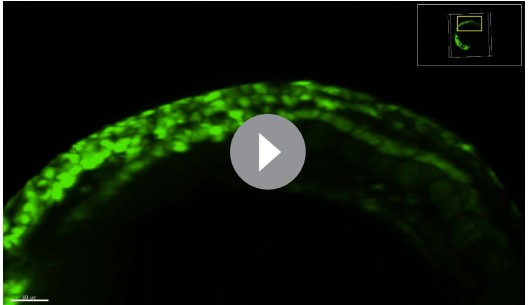

**Video 4.** Neural crest migration in DMSO-treated zebrafish embryos. *Sox10:GFP* transgenic embryos were treated at 2 ss and mounted for imaging at 15 ss. Embryos were imaged for 12 hr, and images were collected every 10 min.
DOI: https://doi.org/10.7554/eLife.29145.035

**Video 5.** Neural crest migration in CAPE-treated (10 µM) zebrafish embryos. *Sox10:GFP* transgenic embryos were treated at 2 ss and mounted for imaging at 15 ss. Embryos were imaged for 12 hr, and images were collected every 10 min.
DOI: https://doi.org/10.7554/eLife.29145.036

## Akt signaling regulates neural crest gene expression in vivo

To confirm that Akt activation plays a role in zebrafish neural crest development, we expressed constitutively active human PTEN-mCherry fusion (PTEN S370A, S380A, T382A, T383A, S385A or PTEN QMA) in *crestin:EGFP* transgenic zebrafish embryos (*Gil et al., 2006*; *Stumpf et al., 2016*). After sorting embryos based on mCherry fluorescence at 24 hpf, we found that PTEN QMA-mCherry decreases both phospho-Akt and *crestin:EGFP* expression (*Figure 7A–C*). We found a similar decrease in *crestin* expression by ISH (*Figure 7D*). The PI3K inhibitor LY294002 also decreased *crestin:EGFP* expression in vivo in a dose responsive manner (*Figure 7E*). Similar to CAPE treatment, we found that co-injection of PTEN QMA prevented *sox10* from stimulating *crestin:EGFP* expression (*Figure 7F–G*). These data support a role for Akt activation in Sox10-dependent neural crest gene transcription.

## Discussion

Neural plate border cells have the capacity to form both neural and non-neural ectodermal derivatives in addition to neural crest cells. While neural plate border specifiers direct expression of neural crest specifiers, the role of additional cues in promoting neural crest specifier expression and activity are poorly characterized. In this study, we used in vitro chemical screening to establish a key role for PI3K/Akt signaling in neural crest development.

Chemical screening in zebrafish provided several advantages over traditional approaches. First, we used an in vitro screening system that facilitated automation, minimized the time-dependency of neural crest gene expression, and maintained cell type heterogeneity. Second, we treated zebrafish embryos at a developmental time point when the neural plate border is already established. Third, we avoided a common limitation of genetic studies in which isoform redundancy masks a phenotype in loss-of-function studies.

We found that CAPE treatment and subsequent inhibition of Akt signaling decreased expression of neural crest genes and led to a failure of neural crest cells to migrate. CAPE also reduced the number of pigmented melanocytes in zebrafish embryos. CAPE-treated embryos exhibited severe defects or death after more than two days of development, confounding analysis of other neural crest derivatives such as peripheral neurons and cartilage. Though CAPE has many reported activities, we found that inhibition of Akt signaling was relevant for its effects on the neural crest. In CAPE-treated embryos, constitutively active Akt1 rescued *crestin* expression, melanocyte number, and melanocyte migration.

We further used neural crest selective cell cultures to investigate CAPE's mechanism of action in the context of particular growth factor stimulation. Although FGF itself did not appear to stimulate Akt phosphorylation, FGF was required for the activity of CAPE. We hypothesized that FGF could activate both PI3K/Akt and Mek/Erk pathways, but that Mek/Erk activation leads to PI3K/Akt cross-inhibition, as has been previously published (*Yu et al., 2002*; *Turke et al., 2012*;

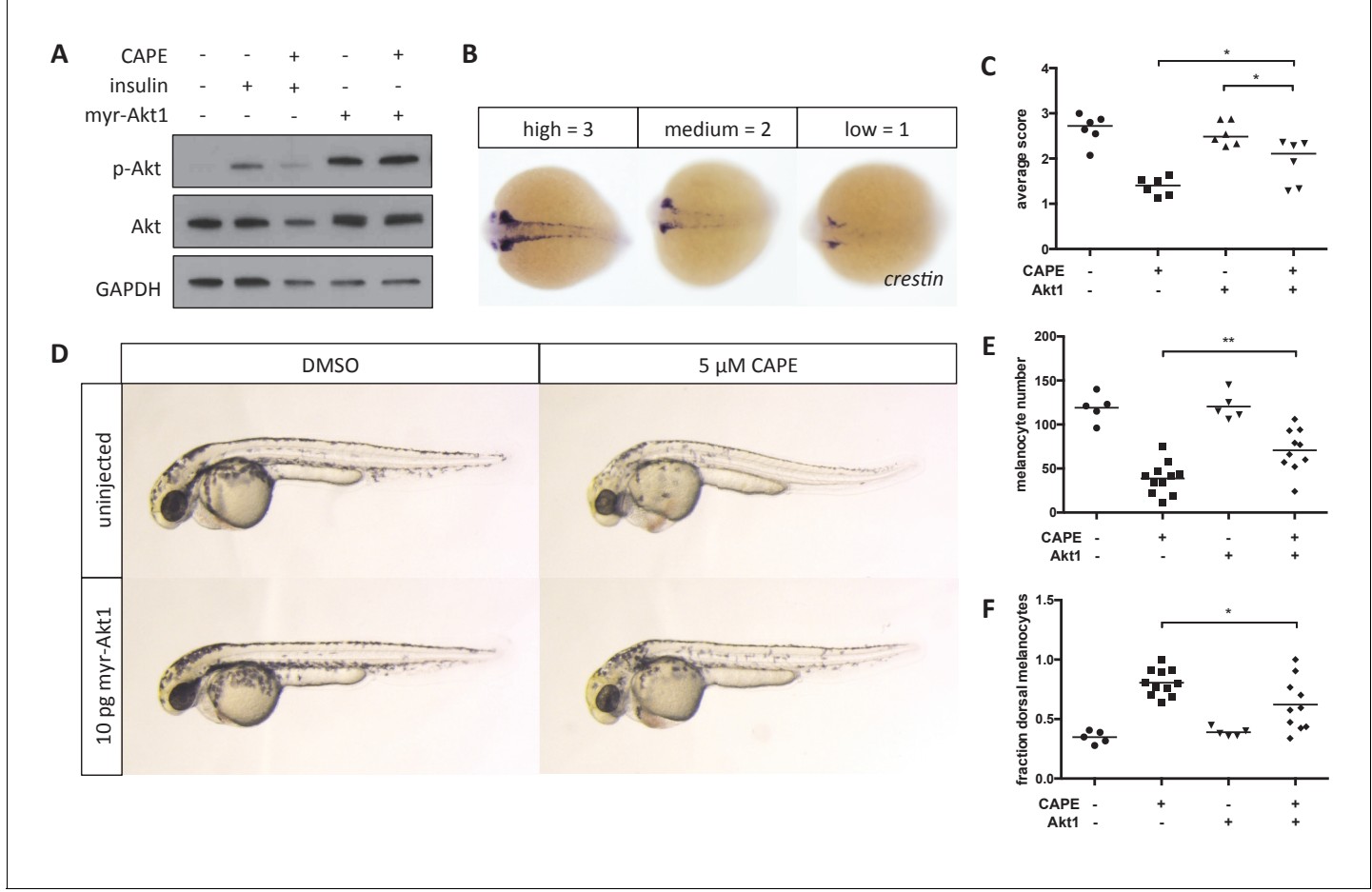

**Figure 5.** Myr-Akt1 rescues neural crest defects caused by CAPE. (A) Injection of myr-Akt1 RNA results in high phospho-Akt in heterogeneous neural crest cultures regardless of CAPE treatment. The same result was observed in four independent experiments. (B) Scoring system for *crestin* in situ hybridization. (C) Myr-Akt1 injection increases *crestin* expression in CAPE-treated embryos. Each point represents the average score of embryos from a single clutch (23–66 embryos per clutch). Three independent experiments are shown. (D) Morphology and pigmentation of CAPE-treated and injected embryos at 2 dpf. Images are representative of three independent experiments. (E) Myr-Akt1 increases melanocyte number in CAPE-treated embryos. Trunk melanocytes were counted as in *Figure 4*. Each point represents one embryo from the same clutch; bars indicate mean. (F) Myr-Akt1 reduces the fraction of dorsal melanocytes in CAPE-treated embryos. *p<0.05, **p<0.005, (C) paired t-test, (E–F) unpaired t-test. *Figure 5—figure supplement 1* shows the effect of PI3K inhibitors on *crestin:EGFP* expression in vitro and the effect of myr-Akt1 on CAPE-induced embryonic defects.

DOI: https://doi.org/10.7554/eLife.29145.037

The following source data and figure supplements are available for figure 5:

**Source data 1.** Scoring of *crestin* expression by ISH in CAPE-treated and myr-Akt1-injected embryos.
DOI: https://doi.org/10.7554/eLife.29145.039

**Source data 2.** Melanocyte numbers in CAPE-treated and myr-Akt1-injected embryos.
DOI: https://doi.org/10.7554/eLife.29145.040

**Source data 3.** Melanocyte numbers in CAPE-treated and myr-Akt1-injected embryos.
DOI: https://doi.org/10.7554/eLife.29145.041

**Figure supplement 1.** Effect of PI3K inhibitors on *crestin:EGFP* expression in vitro and effect of myr-Akt1 injection on CAPE-induced defects.
DOI: https://doi.org/10.7554/eLife.29145.038

**Figure supplement 1—source data 1.** *Ubi:mCherry+* and *crestin:EGFP+* cell numbers in cultures treated with PI-103.
DOI: https://doi.org/10.7554/eLife.29145.042

**Figure supplement 1—source data 2.** *Ubi:mCherry+* and *crestin:EGFP+* cell numbers in cultures treated with GDC0941.
DOI: https://doi.org/10.7554/eLife.29145.043

**Figure supplement 1—source data 3.** Developmental defects in CAPE-treated and myr-Akt1-injected embryos.
DOI: https://doi.org/10.7554/eLife.29145.044

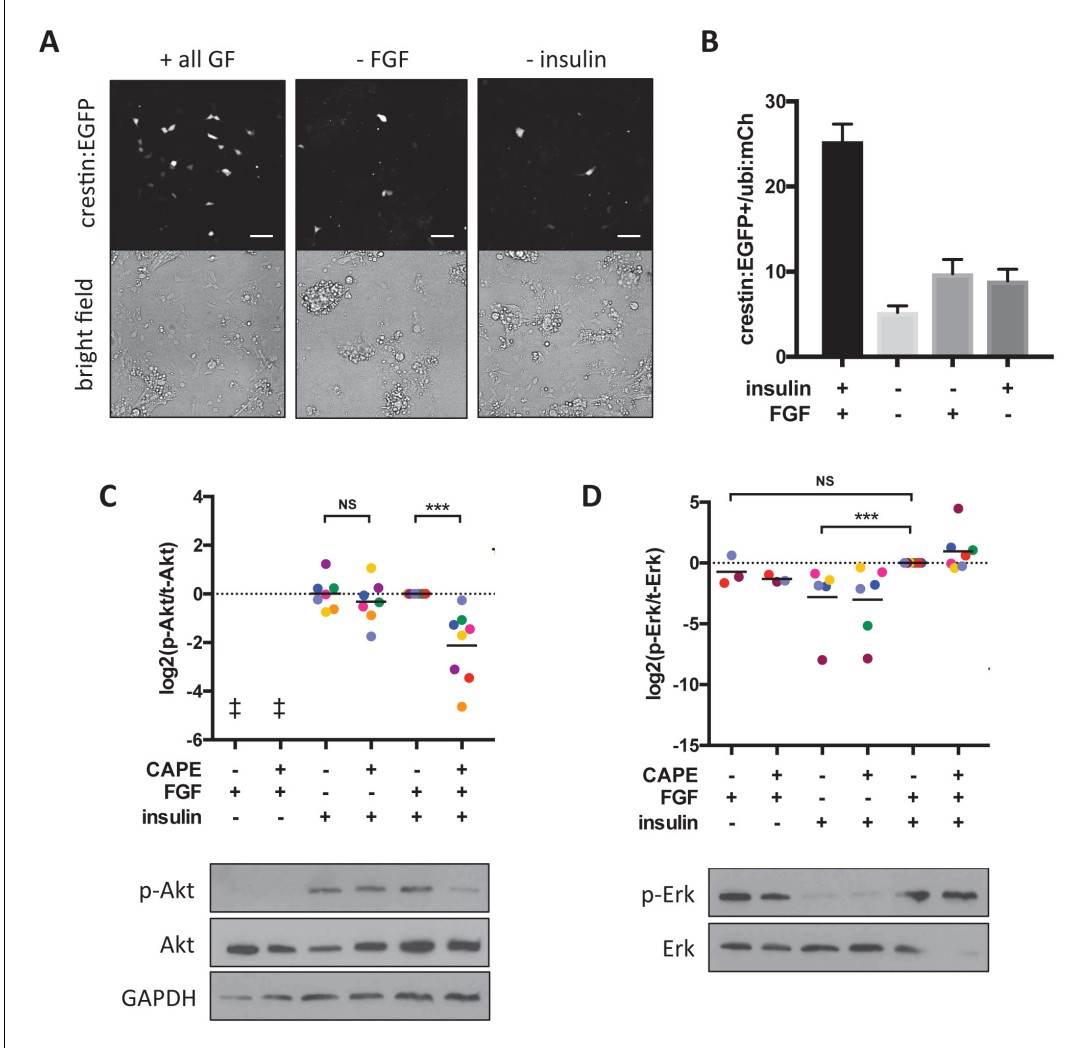

**Figure 6.** CAPE inhibits FGF-stimulated Akt activation in vitro. Embryos were plated in neural crest medium and cultured for 2 hr for western blotting. (A) Heterogeneous neural crest cultures after 24 hr in neural crest medium with or without FGF and insulin. Scale bar: 100 μm. Images are representative of three independent experiments. (B) Quantification of *crestin:EGFP*+ cells for (A). Number of cells was determined by thresholding and normalized to *ubi:mCherry* fluorescence representing total cell number. Mean and standard deviation of at least 4 wells is shown. (C) Ratio of phosphorylated to total Akt. Each point represents an independent experiment corresponding to color. Samples were normalized within an experiment so that p-Akt to Akt ratio with FGF and insulin but without CAPE (condition 5) had a value of 1. Insulin but not FGF stimulation leads to Akt phosphorylation. CAPE inhibits Akt only in FGF-stimulated cells. (D) Ratio of phosphorylated to total Erk. Values were normalized as in (C). FGF but not insulin stimulation leads to Erk phosphorylation. ‡ no signal detected *p<0.05, **p<0.005, ***p<0.0005, paired t-test. *Figure 6—figure supplement 1* shows a model for CAPE's mechanism of action based on the effects of a Mek inhibitor (CI-1040) and a PI3K inhibitor (PI-103) on Akt and Erk phosphorylation. *Figure 6—figure supplement 2* shows the effect of RTK inhibitors on *crestin* expression in vivo.
DOI: https://doi.org/10.7554/eLife.29145.045

The following source data and figure supplements are available for figure 6:

**Source data 1.** Number of *crestin:EGFP+* cells per total *ubi:mCherry* fluorescence.
DOI: https://doi.org/10.7554/eLife.29145.048
**Source data 2.** Quantification of p-Erk/Erk and p-Akt /Akt ratios by western blot and densitometry.
DOI: https://doi.org/10.7554/eLife.29145.049
**Figure supplement 1.** CAPE inhibits FGF-stimulated Akt activation.
DOI: https://doi.org/10.7554/eLife.29145.046
**Figure supplement 1—source data 1.** Scoring of *crestin* expression by ISH in *pten* mutant embryos.
DOI: https://doi.org/10.7554/eLife.29145.050
**Figure supplement 2.** RTK signaling regulates *crestin* expression.
DOI: https://doi.org/10.7554/eLife.29145.047
*Figure 6 continued on next page*

*Figure 6 continued*

**Figure supplement 2—source data 1.** Scoring of *crestin* expression by ISH with kinase inhibitor treatment.
DOI: https://doi.org/10.7554/eLife.29145.051

*Zmajkovicova et al., 2013*; *Menges and McCance, 2008*). This hypothesis was supported by a dramatic increase in Akt phosphorylation upon Mek inhibition (*Figure 6—figure supplement 1A*). We formulated a model in which FGF modulates Akt activation depending on the level of Mek/Erk activity (*Figure 6—figure supplement 1D*). FGF stimulation acts like a pendulum that can swing between PI3K/Akt activation or Mek/Erk activation. Under baseline conditions, FGF stimulation has no net effect on Akt phosphorylation, since FGF both activates Akt and inhibits Akt through Mek/Erk. When a Mek inhibitor is present, FGF stimulates Akt activation. When CAPE is present, FGF-stimulated Akt activation is inhibited, and the negative effect of Mek/Erk on Akt is dominant, leading to decreased Akt phosphorylation. Consistent with this model, the effect of CAPE on Akt phosphorylation was blunted upon Mek inhibition.

The direct target of CAPE is yet to be determined, but CAPE likely acts upstream of PI3K. CAPE does not inhibit phosphorylation of membrane-targeted Akt1, so it acts upstream of PIP3, which recruits Akt to the cell membrane (*Figure 5A*). Hence CAPE could either activate Pten or inhibit PI3K. We ruled out Pten activation by showing that CAPE has an identical effect on *crestin* expression in wild type or *pten* null embryos (*Figure 6—figure supplement 1E*).

Several lines of evidence pointed to Sox10 as a downstream transcription factor affected by CAPE. We found that CAPE treatment decreased chromatin accessibility at Sox10 binding sites, including the promoter of *mitfa*. Furthermore, CAPE's effects on neural crest gene expression mirror that of the zebrafish *colourless* mutant in which *sox10* is disrupted: *sox10* expression is slightly reduced, while expression of *mitfa* and its target genes are dramatically reduced (*Dutton et al., 2001*). The dramatic reduction in both *crestin* and *mitfa* expression in CAPE-treated embryos is consistent with a reduction in Sox10 activity. While Mitf expression depends on both Pax3 and Sox10, only Sox10 binding sites are essential for *crestin* expression, while Pax3 binding sites are dispensable (*Potterf et al., 2000*; *Elworthy et al., 2003*; *Kaufman et al., 2016*). CAPE-treated embryos also show an otic vesicle defect that is similar but not identical to *colourless*. Like *sox10* genetic deficiency, CAPE treatment causes a lack of semicircular canal projects and smaller, more closely spaced otoliths, but it does not decrease otic vesicle size, as has been observed in *sox10* mutants (*Dutton et al., 2009*). Notably, not all phenotypes of *colourless* are recapitulated by CAPE treatment, and CAPE treatment causes defects that are not explained by either Sox10 or Akt deficiency. CAPE likely has other biological targets that explain defects such as body curvature and brain ventricle enlargement. While *colourless* mutants almost entirely lack pigment cells, CAPE primarily affects melanocytes. CAPE may reduce Sox10 activity selectively depending on the target gene, with *mitfa* being one of the most strongly affected genes. One *sox10* mutant, *sox10*[baz1], also has a strong melanocyte defect and weak xanthophore and iridophore defect (*Delfino-Machín et al., 2017*). This mutation causes a single amino acid substitution in the HMG DNA-binding domain of Sox10. Both CAPE and *sox10*[baz1] may selectively affect the activity of Sox10 toward specific target genes, though CAPE's effect could be indirect. CAPE has also been reported to reduce *mitfa* binding to DNA, which could be a secondary mechanism by which CAPE reduces *mitfa* target gene expression in addition to the reduced *mitfa* levels that we observed (*Lee et al., 2013*)

Migration defects in CAPE-treated embryos may be downstream of decreased *mitfa* expression, as in *colourless* mutants, but more direct mechanisms for inhibition of migration downstream of Akt could also play a role. Recent studies have proposed the existence of leader cells with a distinct gene expression profile that initiate neural crest migration (*McLennan et al., 2015*; *Richardson et al., 2016*). Leader cells are reported to depend on PI3K signaling for maintenance of polarity, and PI3K is known to activate Rac, a GTPase important for directed cell migration (*Yamaguchi et al., 2015*). Akt can also directly phosphorylate cytoskeletal filaments and their regulators, resulting in either enhancement or repression of migration depending on context (*Cantley, 2002*). CAPE likely has an additional effect on cell migration beyond the neural crest.

We found that both CAPE treatment and constitutively active PTEN reduced the ability of Sox10 to stimulate *crestin* expression. Sox10 activity might be modified directly, through its binding partners, or through chromatin modification at its binding sites. For example, Akt inhibition could alter

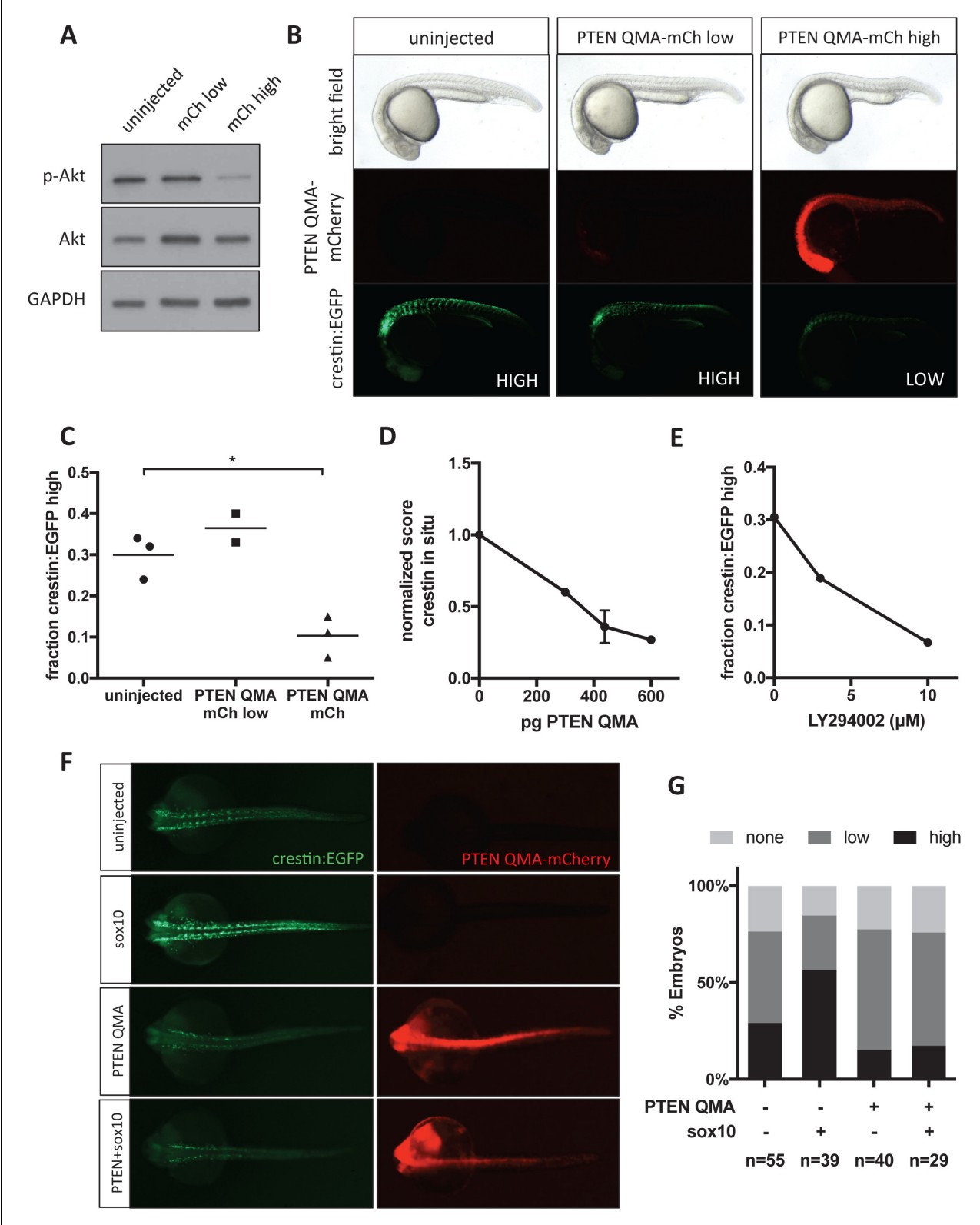

**Figure 7.** Akt signaling regulates neural crest gene expression in vivo. (**A**) PTEN QMA-mCherry (300 pg) reduces phospho-Akt level in whole embryos. (**B**) Morphology, PTEN QMA-mCherry expression, and *crestin:EGFP* expression of PTEN QMA-mCherry injected embryos. Scoring system for *crestin: EGFP* levels is indicated. (**C**) Quantification based on scoring in (**B**). Each point represents a single clutch (5–53 embryos per condition), and mean is indicated. One clutch generated less than 5 PTEN QMA-mCherry low embryos and is not shown. *p<0.01, Student's t-test (**D**) PTEN QMA-mCherry

*Figure 7 continued on next page*

*Figure 7 continued*

decreases *crestin* expression by ISH. Embryos were scored as in **Figure 5B**. Average score normalized to uninjected control is indicated. Error bars represent SEM of four clutches (7–28 embryos per condition) from three independent experiments. One clutch was excluded because of abnormally low staining in both control and injected embryos. (E) The PI3K inhibitor LY294002 decreases *crestin:EGFP* expression. Points indicate the average of two independent experiments. (F) PTEN QMA-mCherry prevents Sox10 from stimulating *crestin:EGFP* expression. Representative embryos from the same clutch are shown. (G) Three clutches (8–19 embryos per condition) from two independent experiments were scored as in (B).

DOI: https://doi.org/10.7554/eLife.29145.052

The following source data is available for figure 7:

**Source data 1.** Scoring of *crestin:EGFP* expression in PTEN QMA-mCherry-injected embryos.
DOI: https://doi.org/10.7554/eLife.29145.053
**Source data 2.** Scoring of *crestin* expression by ISH in PTEN QMA-injected embryos.
DOI: https://doi.org/10.7554/eLife.29145.054
**Source data 3.** Scoring of *crestin:EGFP* expression in LY294002-treated embryos.
DOI: https://doi.org/10.7554/eLife.29145.055
**Source data 4.** Scoring of *crestin:EGFP* in *sox10-* and PTEN-injected embryos.
DOI: https://doi.org/10.7554/eLife.29145.056

Sox10 dimerization or Sox10-Pax3 interaction, both of which are known to affect expression of Sox10 target genes (*Lang et al., 2000*; *Bondurand et al., 2000*; *Potterf et al., 2000*). Akt can directly phosphorylate transcription factors to regulate their activity, as was shown with Twist-1 in tumor cells (*Vichalkovski et al., 2010*). Whether CAPE treatment and Akt inhibition affect Sox10 activity directly or indirectly will be an interesting area for future study.

Our data suggests that CAPE treatment and inhibition of Akt signaling block neural crest differentiation. Interestingly, a prior study in *Xenopus* showed an earlier and broader role for Akt in ectoderm development (*Pegoraro et al., 2015*). Pegoraro et al. found that the glycolysis regulator PFKFB4 is required for specification of all dorsal ectoderm derivatives, including neural and non-neural ectoderm, neural crest, and placodes, and acts by inhibiting Akt signaling. In contrast, our study found that inhibition of Akt signaling does not disrupt early neural crest specification but inhibits subsequent expression of Sox10 target genes, neural crest migration, and pigment cell differentiation. This apparent discrepancy could be the result of species-specific differences, or it could be related to the specificity of CAPE for inhibition of Akt stimulated by a particular ligand. We showed that in vitro CAPE inhibits FGF-stimulated Akt activation but not insulin-stimulated Akt activation. Though our study and that of Pegoraro et al. suggest different roles for Akt signaling in neural crest development, they both demonstrate that Akt signaling promotes differentiation independently of its effect on cell survival. A current challenge in neural crest biology is understanding how morphogens connect to gene expression through intracellular signaling pathways and their associated transcription factors. Our study has identified PI3K/Akt as a novel intracellular signaling pathway that regulates neural crest differentiation through modulation of Sox10 activity. Our findings may also impact the understanding and treatment of melanoma, as the Sox10 target *MITF* is a lineage oncogene in this neural crest-derived cancer (*Garraway et al., 2005*).

## Materials and methods

### Zebrafish husbandry and strains

Zebrafish were maintained under standard protocols approved by the Boston Children's Hospital (BCH) Institutional Animal Care and Use Committee (IACUC). Unless indicated otherwise, AB strain embryos were used for all experiments. Additional zebrafish strains included *crestin_1 kb:EGFP* (referred to as *crestin:EGFP* here) (*Kaufman et al., 2016*) (RRID:ZFIN_ ZDB-TGCONSTRCT-160208–1), *crestin_296bp:EGFP* (*Kaufman et al., 2016*) (RRID:ZFIN_ ZDB-TGCONSTRCT-160208–2), *sox10: GFP* (*Curtin et al., 2011*) (RRID:ZFIN_ ZDB-ALT-110411–1), *sox10:Kaede* (*Dougherty et al., 2012*) (RRID:ZFIN_ ZDB-FISH-150901–26827), *mitfa:GFP* (*Curran et al., 2009*) (RRID:ZFIN_ ZDB-FISH-150901–12193), *ubi:mCherry* (*Mosimann et al., 2011*) (RRID:ZFIN_ ZDB-ALT-110317–3), and *casper* (*White et al., 2008*) (RRID:ZFIN_ ZDB-FISH-150901–6638, *ptena-/-ptenb-/-* (*Faucherre et al., 2008*) (RRID:ZFIN_ ZDB-FISH-150901–12142). Zebrafish embryos were staged according to (*Kimmel et al., 1995*).

## Heterogeneous neural crest cell culture

*Crestin:EGFP; ubi:mCherry* transgenic zebrafish embryos were grown to the 5 somite stage, decontaminated in 0.05% bleach for 2 min, and dechorionated in 2.5 mg/ml pronase for 5 min. Embryos were then mechanically homogenized in neural crest medium using a rotor-stator homogenizer (Omni TH) at 5000 rpm for 10 s. Neural crest medium consisted of a base medium of DMEM/F12 with L-glutamine and 2.438 g/l sodium bicarbonate (Invitrogen #11320) supplemented with 12% FBS, 1% N2 supplement (Invitrogen #17502), 20 μg/ml insulin (Gemini Bioproducts #700–112P), 20 ng/ml FGF2 (R and D Systems #233-FB-025/CF), 20 ng/ml EGF (R and D Systems #236-EG-200), and 0.2% Primocin (InVivoGen). Embryos were plated at a density of 0.6–1 embryos per 0.1 cm$^2$ (1 well of a 384 well plate) on standard tissue culture-coated plates and cultured for 24–48 hr under 5% $CO_2$ at 28.5°C. N2 was excluded for short term (2 hr) cultures to test the effects of growth factor stimulation since it contains insulin.

## In vitro immunofluorescence, EdU staining, and migration speed measurement

EdU staining was conducted using the Click-iT EdU Flow Cytometry Assay Kit (Invitrogen). Zebrafish embryonic cells were plated on collagen, allowed to attach for 24 hr, and treated with 10 μM EdU for 4 hr. Cells were then fixed in 2% PFA for 30 min at RT, permeabilized in 0.5% Triton-X100/PBS, and blocked in 5% lamb serum in 0.1% Triton X-100/PBS for 1 hr at RT. Cells were incubated with primary antibody (AnaSpec anti-GFP #55423, 1:500) in blocking buffer overnight. Cells were washed 4 times and incubated for 2 hr at RT in secondary antibody (Alexa Fluor 488 goat anti-chicken, 1:2000). Cells were washed once, and the Click-iT reaction was performed with Alexa Fluor 647 azide. Cell speed was determined using time lapse fluorescence microscopy of *crestin:EGFP+* and *ubi:mCherry+* cells with images acquired every 2 min. The center of each cell was manually tracked using ImageJ over a period of 4 hr, and the sum of distances migrated per frame was divided by total imaging time.

## Quantitative PCR

RNA was isolated using the RNeasy Plus Mini Kit (Qiagen) according to the manufacturer's instructions. RNA was quantified by absorbance at 260 nm and a standard amount was used as input to an iScript reverse transcriptase reaction (Bio-Rad). The product of this reaction was used for quantitative PCR at a maximum volume of 1 μl per 10 μl final reaction volume. Quantitative PCR was conducted using SsoFast EvaGreen Supermix (Bio-Rad) and run on a CFX384 Real-Time System/C1000 Thermal Cycler (Bio-Rad). Reactions were conducted in triplicate and normalized to β-actin. For each gene, the sample with the highest expression was assigned a value of 1 and other samples were normalized accordingly. Primer sequences are displayed in *Table 2*.

## Intradermal cell transplantation

Heterogeneous neural crest cell cultures were sorted for DAPI negative, crestin:EGFP+ cells and resuspended in PBS at a concentration of 10,000 cells/μl. Cell suspension was loaded into a glass syringe with a 33 gauge needle (Hamilton), and 0.5 μl was injected underneath a scale on the dorsal flank of a *casper* fish irradiated with 15 Gy per day for two consecutive days. Transplants were conducted on the third day.

## Chemical screening and hit determination

Embryos were plated with chemicals at 5 ss in 384-well plates as described in heterogeneous NC cultures. Cells were cultured for 1 day, Hoechst 33342 was added at a final concentration of 0.3 μg/ml, and wells were imaged in red, green, and blue channels on a Nikon Eclipse Ti Spinning Disk Confocal. Chemicals libraries included LOPAC1280, a custom bioactives library, FDA-approved drugs, and Chembridge KINAcore and NHRcore libraries. Chemicals were screened in triplicate at two concentrations depending on the library. Hits were identified based on deviation from plate controls in two values: ratio of *crestin:EGFP+/ubi:mCherry+* cells and *crestin:EGFP+/total nuclei*. All hits were verified by eye.

**Table 2.** Primer sequences.

| Use | Gene | Forward | Reverse | Reference |
|---|---|---|---|---|
| qPCR | bactin1 | CGAGCAGGAGATGGGAACC | CAACGGAAACGCTCATTGC | (*McCurley and Callard, 2008*) |
| qPCR | sox10 | ATATCCGCACCTGCACAA | CGTTCAGCAGTCTCCACAG | |
| qPCR | crestin | AGTGCCTGCCAATGTTCAC | CTGAAAAAGGCCGATGAGTT | |
| qPCR | foxd3 | CATGCAAAACAAGCCCAAG | ATGAGGGCGATGTACGAGTAG | |
| qPCR | mitfa | GGCGGTTTAATATCAATGACAGA | GGTGCCTTTATTCCACCTCA | |
| qPCR | neurog1 | CGTGCCATTATCTTCAACACA | CGATCTCCATTGTTGATAACCTT | |
| qPCR | myf5 | GCTACAACTTTGACGCACAAA | CACGATGCTGGACAAACACT | |
| qPCR | runx1 | CGTCTTCACAAACCCTCCTCAA | GCTTTACTGCTTCATCCGGCT | |
| ISH | tfap2a | TAATACGACTCACTATAGGGAATCT TCACAGATGTTAGTGCACAGTTTTTCCGCGAT | AATTAACCCTCACTAAAGGTCAC TTTCTGTGCTTCTCATCTT | |
| ISH | tfap2c | TAATACGACTCACTATAGGGACAG AAACAACATGTTGTGGAAATTAGCAGATAA | AATTAACCCTCACTAAAGGTCA CTTTCGGTGTTTGTCCATCTT | |
| ISH | inka1a | AATTAACCCTCACTAAAGGG GAATCGGGTGACTGTCTGC | TAATACGACTCACTATAGGGATGG GTGTTCTGCTCCCAG | |
| ISH | dlx2a | AATTAACCCTCACTAAAGGACAA CAGCATGAACAGCGTC | TAATACGACTCACTATAGGGACAGGC GCATGAAACACAT | |
| ISH | pax7a | AATTAACCCTCACTAAAGGAGAA CTACCCACGAACCGGA | TAATACGACTCACTATAGGTTGATC TGTGAAGCGTGCTG | |
| ISH | myca | TAATACGACTCACTATAGGGCAAG TGTCAAAATGCCGGTGAGTGCGAGTTTGGCGT | AATTAACCCTCACTAAAGGTTAATGTG AACTCCGCAGCTGCTGAA | |
| ISH | ets1 | TAATACGACTCACTATAGGGTGTA CGTTTGAATGCGTGACCATGACGGCAGCTGT | AATTAACCCTCACTAAAGGTCAGGAGC TCCAACAGGAACTGCCAGA | |
| ISH | nr2f2 | TAATACGACTCACTATAGGGTAGATATGGC AATGGTAGTGTGGAGAGGCTCCCA | AATTAACCCTCACTAAAGGCTACTGAAT CGACATATAAGGCCAGTT | |
| ISH | msx1b | TAATACGACTCACTATAGGGGATGGTTAA CGATGAATTCTCCTAAGGGACCCGTT | AATTAACCCTCACTAAAGGTTAAGAC AAATAATACATCCCATA | |
| ISH | dlx5a | TAATACGACTCACTATAGGGTTATCCAA ACTATGACTGGAGTATTCGACAGAAGGA | AATTAACCCTCACTAAAGGTCAGTACAAC GTTCCTGATCCGAGTGCCAA | |

DOI: https://doi.org/10.7554/eLife.29145.057

## Chemical treatment of zebrafish embryos

Unless otherwise indicated, embryos were treated at 2 ss. To reach this time point, embryos fertilized in the morning were either incubated at 28.5°C for 12 hr or 23°C for the first 6 hr of development, then transferred to 19°C overnight. Embryos fertilized in the afternoon were incubated at 23°C overnight. Chemical stocks were maintained in DMSO, resulting in a final concentration of no more than 0.3% DMSO in E3 embryo medium. Embryos at a maximum density of 20/well were treated in 24 well plates in 1 ml of chemical solution at 28.5°C protected from light. CAPE was obtained from Tocris (cat #2743) and used at a concentration of 10 µM unless otherwise indicated.

## Fluorescence activated cell sorting

Cell sorting was conducted on a BD FACSAria IIu using a nozzle diameter of 80 µm. For whole embryo FACS, embryos were mechanically homogenized in FACS buffer (2% FBS in PBS), passed through a 40 µm cell strainer and kept on ice until sorting. DAPI was added at a final concentration of 1 µg/ml to distinguish live from dead cells. Cells were collected in media or FACS buffer and kept on ice until further analysis. For *crestin:EGFP* sorting, the ratio of GFP to PE signal was used to distinguish autofluorescence from GFP fluorescence.

## Imaging of zebrafish embryos

Epifluorescence and bright field images were obtained using a Zeiss Discovery V.8 Stereoscope with an Axiocam HRc. Confocal microscopy of flat-mounted embryos was conducted on a Nikon C2si Laser Scanning Confocal. Time lapse confocal microscopy of live embryos was conducted on a Nikon

Eclipse Ti Spinning Disk Confocal. Embryos were mounted on 6 well imaging plates in 0.8% low melting point agarose containing 0.003% 1-phenyl-2-thiourea (PTU) to prevent pigmentation, 1.6 µg/ml tricaine (MS-222) to immobilize fish, and the relevant concentration of CAPE. Embryos were maintained at 28.5°C during imaging.

## Whole mount in situ hybridization, immunofluorescence, and TUNEL

Whole mount in situ hybridization was conducted as described (*Tallquist and Soriano, 2003*; *Thisse and Thisse, 2008*; *Liu et al., 2002*). The following probes were generated from established plasmids: *crestin* (*Rubinstein et al., 2000*), *pax3* (*Seo et al., 1998*), *foxd3*, *sox10*, and *snai1b* (*Thisse et al., 1995*). Plasmid templates were linearized using an appropriate restriction enzyme and PCR purified prior to in vitro transcription. Other in situ probes were generated from PCR products containing a T7 promoter: *tfap2a*, *tfap2c*, *inka1a*, *dlx2a*, *pax7a*, *myca*, *ets1*, *nr2f2*, *msx1b*, *dlx5a*. See *Table 2* for primer sequences. RNA was isolated from 17 ss embryos using the RNeasy Plus Mini Kit (Qiagen). A cDNA library was synthesized using the SuperScript III First Strand Synthesis System (Invitrogen). PCR was conducted using Phusion High-Fidelity DNA Polymerase (NEB), and products were purified prior to in vitro transcription with the following components: Roche Dig 11277073910, BCIP/NBT S3771, Roche T3 or T7 polymerase, RNaseIn (Promega).

For whole mount immunofluorescence, zebrafish embryos were fixed in 4% PFA overnight at 4°C. Embryos were permeabilized in acetone at −20°C for 7 min, blocked for 30 min in blocking buffer (2% BSA, 10% lamb serum, 1% DMSO in PBS-0.1% Triton X-100) and incubated in primary antibodies in blocking buffer overnight. Embryos were washed twice for 30 min in PBS-0.1% Triton X-100, then incubated for 2 hr with secondary antibodies at room temperature. Anti-phospho-H3 antibody was obtained from Santa Cruz Biotechnology (rabbit polyclonal, sc-8656-R, 1:750) and anti-GFP was from Genetex (chicken polyclonal, GTX13970, 1:500). Secondary antibodies were from Thermo Fisher (goat anti-rabbit IgG Alexa Fluor 555, goat anti-chicken Alexa Fluor 488, both 1:1000).

Whole mount TUNEL was conducted after *crestin* in situ hybridization using the ApopTag Peroxidase In Situ Apoptosis Detection Kit (EMD Millipore) with extended incubation times. Signal was visualized with SIGMAFAST 3,3-Diaminobenzidine tablets (Sigma).

## RNA-seq

Total RNA was extracted from sorted *sox10:GFP+* cells using Trizol LS according to the manufacturer's instructions. Libraries were prepared using the Ribogone kit (Clontech) and the SMARTer Universal Low RNA Kit (Clontech) according to the manufacturer's instructions. Libraries were analyzed on a Fragment Analyzer (Advanced Analytical) and quantified using the Qubit (Invitrogen) prior to sequencing on the Illumina HiSeq 2500. Quality control of RNA-Seq datasets was performed by FastQC and Cutadapt to remove adaptor sequences and low quality regions. The high-quality reads were aligned to UCSC build danRer7 of the zebrafish genome using Tophat 2.0.11 without novel splicing form calls. Transcript abundance and differential expression were calculated with Cufflinks 2.2.1. FPKM values were used to normalize and quantify each transcript.

## Plasmids

Myr-Akt1 was a gift from William Sellers (Addgene plasmid #9008) (*Ramaswamy et al., 1999*). pCSDest was a gift from Nathan Lawson (Addgene plasmid # 22423). PTEN QMA-mCherry was from *Stumpf et al., 2016*.

## Western blotting

Cultured cells were collected on ice, pelleted at 500 rcf for 3 min at 4°C, washed once with ice-cold PBS, and lysed in RIPA buffer containing protease and phosphatase inhibitors. Embryos were dechorionated, deyolked on ice in 55 mM NaCl, 1.8 mM KCl, 1.3 mM NaHCO₃, centrifuged at 500 rcf for 3 min at 4°C, and washed once in 110 mM NaCl, 3.5 mM KCl, 2.7 mM CaCl₂, 10 mM Tris pH 8.5 prior to lysis as for cultured cells. Lysate was centrifuged at 20,000 rcf for 10 min and the supernatant was collected for analysis of protein concentration with the DC Protein Assay (Bio-Rad). Samples were boiled in Laemelli buffer before loading of 10 µg protein per well of a 4–20% polyacrylamide gel. Protein was transferred to a PVDF membrane using the iBlot dry blotting system (Invitrogen), blocked in 5% milk in TBS with 0.1% Tween 20, and incubated in primary antibody in

5% BSA overnight at 4°C. Blots were washed, incubated in horse radish peroxidase (HRP)-conjugated secondary antibody for 1 hr at room temperature, washed again, and developed using Amersham ECL Prime Western Blotting Detection Reagent (GE Healthcare) or Pierce ECL Western Blotting Reagent (Thermo Fisher Scientific). Antibodies were obtained from Cell Signaling: Erk (#9102, RRID:AB_330744, 1:1000), Akt (#9272, RRID:AB_329827, 1:1000), phospho-Akt (#9271, RRID:AB_329825, 1:1000), phospho-Erk (#9101, RRID:AB_331646, 1:1000), GAPDH (#2118, RRID: AB_561053, 1:2000), HRP-linked anti-rabbit IgG (#7074, RRID:AB_2099233, 1:2000).

## RNA microinjection

Constructs were cloned into pCSDest (*Villefranc et al., 2007*), linearized, transcribed using the Ambion mMessage mMachine SP6 Kit (Thermo Fisher Scientific), and purified with the RNeasy MinElute RNA Cleanup Kit (Qiagen). RNA was quantified based on absorbance at 260 nm and injected with 0.1% phenol red into 1–2 cell embryos in a volume of 1–2 nl.

## ATAC-seq

Cells (12,000–40,000) were pelleted at 500 rcf for 5 min at 4°C, washed once in ice cold PBS, and permeabilized in 10 mM Tris-HCl pH 7.4, 10 mM NaCl, 3 mM MgCl2, 0.1% IGEPAL CA-360. Libraries were prepared using the Nextera DNA Preparation Kit (Illumina) with a transposase reaction time of 30–45 min and purified with the MinElute PCR Purification Kit (Qiagen). Quantitative PCR was used to estimate the total number of cycles needed for library amplification. Libraries were analyzed on a Fragment Analyzer (Advanced Analytical) and quantified using the Qubit (Invitrogen) prior to sequencing on the Illumina HiSeq 2500. ATAC-Seq datasets were aligned to UCSC build danRer7 of the zebrafish genome using Bowtie2 (version 2.2.1) with the following parameters: –end-to-end, -N0, -L20. We used the MACS2 version 2.1.0 peak finding algorithm to identify regions of ATAC-Seq peaks, with the following parameter –nomodel –shift −100 –extsize 200. A q-value threshold of enrichment of 0.05 was used for all datasets. HOMER was used for peak motif analysis (RRID:SCR_010881).

## Acknowledgements

We would like to thank Dipti Gupta and Manav Gupta for assistance with screening robots, Ilya Shestopalov and Elliott Hagedorn for imaging assistance, Robert Mathieu and Mahnaz Paktinat for FACS at the BCH Flow Cytometry Core, and Kristin Artinger for critical review of the manuscript.

## Additional information

### Competing interests

Leonard I Zon: L.I.Z. is a founder and stock holder of Fate Therapeutics, Marauder Therapeutics, and Scholar Rock. The other authors declare that no competing interests exist.

### Funding

| Funder | Grant reference number | Author |
| --- | --- | --- |
| National Institutes of Health | F31CA180313 | Christie Ciarlo |
| Melanoma Research Alliance | | Leonard I Zon |
| Lawrence Ellison Foundation | | Leonard I Zon |
| Howard Hughes Medical Institute | | Leonard I Zon |
| National Institutes of Health | R01CA103846 | Leonard I Zon |
| National Institutes of Health | RO3DE024490 | Eric Liao |
| National Institutes of Health | K08AR061071 | Charles K Kaufman |

The funders had no role in study design, data collection and interpretation, or the decision to submit the work for publication.

## Author contributions
Christie Ciarlo, Conceptualization, Formal analysis, Investigation, Visualization, Methodology, Writing—original draft; Charles K Kaufman, Sasja Blokzijl-Franke, Resources, Investigation; Beste Kinikoglu, Investigation, Methodology; Jonathan Michael, Investigation, Conducted and analyzed experiments including western blotting and in situ hybridization; Song Yang, Formal analysis, Analyzed RNA-seq and ATAC-seq data; Christopher D'Amato, Investigation, Conducted and analyzed experiments including zebrafish primary embryonic cell culture and in situ hybridization; Jeroen den Hertog, Resources, Supervision, Investigation; Thorsten M Schlaeger, Resources, Methodology; Yi Zhou, Formal analysis, Supervision; Eric Liao, Supervision, Funding acquisition, Investigation; Leonard I Zon, Conceptualization, Supervision, Funding acquisition, Investigation, Writing—review and editing

## Author ORCIDs
Christie Ciarlo [iD] https://orcid.org/0000-0002-2876-2432
Leonard I Zon [iD] https://orcid.org/0000-0003-0860-926X

## Ethics
Animal experimentation: Zebrafish were maintained under standard protocols approved by the Boston Children's Hospital (BCH) Institutional Animal Care and Use Committee (IACUC) (protocol # 14-10-2789R).

## Decision letter and Author response
Decision letter https://doi.org/10.7554/eLife.29145.064
Author response https://doi.org/10.7554/eLife.29145.065

# Additional files

## Supplementary files
• Transparent reporting form
DOI: https://doi.org/10.7554/eLife.29145.058

## Major datasets
The following dataset was generated:

| Author(s) | Year | Dataset title | Dataset URL | Database, license, and accessibility information |
|---|---|---|---|---|
| Ciarlo C, Kaufman CK, Kinikoglu B, Michael J, Yang S, D'Amato C, Blokzijl-Franke S, den Hertog J, Schlaeger TM, Zhou Y, Liao EC, Zon LI | 2017 | A chemical screen in zebrafish embryonic cells establishes that Akt activation is required for neural crest development | https://www.ncbi.nlm.nih.gov/geo/query/acc.cgi?acc=GSE95815 | Publicly available at the NCBI Gene Expression Omnibus (accession no: GSE95815) |

The following previously published dataset was used:

| Author(s) | Year | Dataset title | Dataset URL | Database, license, and accessibility information |
|---|---|---|---|---|
| Kaufman CK, Mosimann C, Fan ZP, Yang S, Thomas AJ, Ablain J, Tan JL, Fogley RD, van Rooijen E, Hagedorn EJ, Ciarlo C, White RM, Matos DA, Puller AC, Santoriello C, Liao EC, Young RA, Zon LI. | 2016 | A zebrafish melanoma model reveals emergence of neural crest identity during melanoma initiation | https://www.ncbi.nlm.nih.gov/geo/query/acc.cgi?acc=GSE75356 | Publicly available at the NCBI Gene Expression Omnibus (accession no: GSE75356) |

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
