## [Decision Letter]

[Editors’ note: a previous version of this study was rejected after peer review, but the authors submitted for reconsideration. The first decision letter after peer review is shown below.]

Thank you for submitting your work entitled "A chemical screen in zebrafish embryonic cells establishes that Akt activation is required for neural crest development" for consideration by *eLife*. Your article has been reviewed by three peer reviewers, one of whom is a member of our Board of Reviewing Editors, and the evaluation has been overseen by a Senior Editor. The reviewers have opted to remain anonymous.

Our decision has been reached after consultation between the reviewers. Based on these discussions and the individual reviews below, we regret to inform you that your work will not be considered further for publication in *eLife* at the present time. The reviewers all found the study interesting and potentially important, and highlighted the design of the screening assay as a strength of the study. However, there were concerns over the specificity of some of the experimental approaches, quantitation of the data, and degree of advance and mechanistic insight over previously published work. The reviewers have identified a list of revisions, but it was felt that it would be unrealistic to expect that these could be completed within a timeframe of two months.

As it is the policy of *eLife* to only encourage resubmission of papers that can modified within a relatively short timeframe, I have no choice but to decline the present version of the paper. If in the future you feel you can address the detailed comments of the reviewers, we would be open to receiving a new version of the manuscript and make every effort to the paper to the original reviewers.

Reviewer #1:

This is an interesting study that has identified compounds that interfere with neural crest development using an in vitro-based screening assay followed by validation in vivo in the zebrafish embryo. The study focusses on one compound, CAPE, which shows a dose-dependent reduction in sox10 activity. The authors provide evidence to show that CAPE mediates its effects through inhibition of PI3K/Akt signalling in FGF-stimulated cells, resulting in abnormal neural crest development and some other embryonic defects. The study uses a range of experimental approaches to support the main conclusions. I have a few suggestions for improved clarity or presentation of the data.

1) Although the main conclusion is that CAPE acts by reducing Sox10 activity, it should be pointed out more explicitly that the phenotype of CAPE-treated embryos does not resemble that of sox10 mutants closely. The neural crest defects (pigmentation) are less severe, while other aspects (e.g. body curvature and length, pericardial oedema, small head and eyes etc) are more severe. The current text only highlights similarities. The authors point out that CAPE only appears to affect sox10 activity at some promoters, but I think it would be helpful to have show a more detailed comparison with the known effects of the genetic loss of sox10 function. For example, sox10 mutants lack all three pigment cell types, not just melanophores. Are all three pigment cell types reduced in the CAPE-treated embryos, or does treatment only affect the melanophores? (Clearly, some xanthophores are still present, given the yellow colouration of the head in CAPE-treated embryos.) Iridophores should also be checked, since these are missing in sox10 mutants, but are increased at the expense of melanophores in mitfa mutants (Lister et al., 1999).

2) The other major site of sox10 expression in the embryo is the ear, and sox10 mutant embryos are known to have a very strong ear phenotype (Dutton et al., 2009). Does CAPE treatment also recapitulate these otic defects? If yes, this would support the interpretation that CAPE acts via Sox10, and if no, it might reveal interesting differences in target gene activity for Sox10 between the ear and the neural crest.

3) With respect to the effects on the embryo such as body curvature, which are unlikely to be attributed to a loss of Sox10 activity, what is the interpretation? Are there any more general effects on cell behaviour? The video showing reduced cell migration is clear, but is there a general inhibition of cell migration? What happens to other migratory cell types e.g. the lateral line primordium?

4) The HOMER analysis (Figure 3) is supposed to identify transcription factors in an unbiased manner, and clearly has generated a lot of data. Only the data for the Sox10 and Mitf motifs are shown, though. It is stated that these are the most enriched, but it would be helpful to show how these compare to say, the next three most enriched in the dataset – otherwise this just looks like confirmation of a result that was already predicted.

5) It is not clear where the two PI3K inhibitors (described as 'strong hits') appear in Table 1 – please highlight these. PI3K inhibition is not mentioned for any of the compounds in the Target column. More generally, it would help if this Table conveyed more information. For example, do the hits fall into clear groups based on structural similarities?

Reviewer #2:

This is a very interesting manuscript in which the authors have identified a specific role for pAKT in the progressive acquisition of NC status. The authors here report another success from an ingenious screen, using a mixture of embryonic cells from early stages of development in which NC have been specified, and remain sensitive to modulation. They identify CAPE as a molecule able to inhibit pAKT when NC development is under way (specified). This inhibition of AKT leads to clear defects in the progression of NC development through alteration of expression of NC genes, reporters, melanocyte formation, and migration. Furthermore, AKT inhibition seems to specifically alter SOX10 activity, which in turn mediates at least some of the other NC effects noted.

Their exploration of putative pathway contributions reveals that while CAPE effects on AKT and NC development rely on FGF2 (and perhaps pERK), alternative inhibition of pAKT through other molecules, does lead to AKT inhibition and NC deficits independently of FGF2. This is a solid work with novel insights of signaling that should be very welcomed by the developmental biologists and the neural crest community alike. However a few important concerns are pointed below:

The description made for the IH results lacks a) temporal context of expression of specific genes, and b) a proper quantification. The result comments are qualitative and subjective, (for example the authors suggest a robust change on SOX10 via IH, which is and also lack spatiotemporal context. Are early genes spared alterations triggered by CAPE? Does CASPE only alters later NC markers?, all later NC markers? The RNAseq data from sox10:Kaede+ provides quantifiable data, this suggest minimal effect on Sox10, yet significant changes in other genes (Pax3). A better analysis of the regional pattern for IH along with possible limitations of the transgenic line (does it collect all NC?, could subpopulations be missed by this line?) seems warranted, and will be particularly interesting given the ATAC analysis was also performed on this population.

While describing changes in expression based on IH, the authors bundle SOX10 Crestin and Pax7 as being all dramatically reduced and place other genes like FoxD3, Snai1b, or Inka1a as regionally affected. It is not clear from the images provided that SOX10, or even crestin, are not differentially affected by region. Pax7 instead does seem to be eliminated in normal NC associated regions. In the other hand genes like Pax7, Pax3 and Ets1 seem to increase in Non-NC territories. Additionally, the authors state that genes like Pax3 and Ets1 do not change their profiles in NC territory, yet some of the images provided do not allow a clear conclusion.

It is worth noting that the authors suggest that CAPE and AKT play a role in late events, yet no tests for earlier effects were performed.

While addressing cell death and proliferations the authors quantified effects over region when they have the opportunity to specifically address changes over GFP+ neural crest. While the authors suggest that the dramatic effects in cell death arise late, it seems relevant that the cell death increased at 10µM CAPE, the same concentration used through several other experiments.

The pathway analysis results regarding signaling ligands are not very informative in their current format due to over and under expression of the same pathways. And would suggest to remove this portion.

The Materials and methods section reports that embryonic cell cultures medium contains 12% FBS and N2 (Insulin 5µg/ml), however they do not provide any arguments or considerations regarding these components while addressing the effect of specific signals at multiple points. Did they remove the N2 while addressing FGF2 or Insulin? Certainly FBS has potential interfering components that should have been noted.

It might be relevant to test if other AKT inhibitors (Pi-103 Figure 6—figure supplement 1) could inhibit pAKT triggered by Insulin, FGF2 and iMEK.

The ERK-PI3K analyses were done in zebrafish cell cultures and report broad activity and effects of molecules on signaling. They may not represent signaling effects mediated by NC.

Regarding the model provided in Figure 6 panel E, while ERK activation might block PIP3, this is context dependent. In the experiments here addressed, it seems that ERK inhibition does boost pPI3K, however under Insulin FGF2 activity, robust pERK does not seem to reduce pAKT, and thus the solid line of inhibition between Erk and PI3P might be misleading.

Reviewer #3:

The authors have carried out a chemical screen to identify compounds that reduce expression crestin:gfp in a mixed culture of primary cell derived from dissociated transgenic zebrafish embryos. They focus on one compound, CAPE, and find that it reduces expression of neural crest markers in zebrafish, a few hours before leading to widespread cell death. They provide evidence that CAPE acts upstream of phospho-Akt, a well-known regulator of cell viability.

Mechanistic insight from the manuscript is very limited. I would like to see more depth at one level or another. For instance, it is unclear where CAPE acts. The authors mention a paper in which CAPE was shown to inhibit PI3K activity. Is this the authors' preferred model for the principal locus of control of CAPE in the control of gene expression in NC? If so, why does it depend on FGF signaling? The proposed requirement for FGF signaling is not entirely convincing with the data shown. Figure 6 is rather baffling. CAPE is said to be ineffective in the absence of FGF. But even in the absence of added FGF there must be some basal level of RAS/RAF/ERK/MEK signaling, because addition of a MEK inhibitor massively up regulates the amount of phospho-AKT. The bar chart suggests there is some effect of CAPE, but it does not reach significance. Does constitutively active PI3K rescue the effects of CAPE? This seems unlikely because CA-Akt only partially does so, and loss of pten LOF does not seem to do so. [As an aside, the pten mutant data is buried and is under-explained (Figure 6—figure supplement 1). Comparing Figure 6—figure supplement 1, to Figure 3 makes is appear that ptena -/-; ptenb +/- have an elevated level of crestin in comparison to wild-types. Is that true? Do the double mutants have other phenotypes accompanying the elevated crestin?] Returning to the question of the locus of CAPE activity, does CAPE work to reduce p-AKt levels in ptena/b double mutants? Do the PI3K inhibitors work to rescue elevate crestin in such double mutants? If not, then perhaps PI3K really is the central locus of control. There is more to learn from this mutant that would help point to the locus of CAPE action.

Another option for mechanistic insight that would elevate the significance of the study would be to dig deeper into the transcriptional regulation. What causes the chromatin to close at the MITF promoter? As the authors noted, Β-catenin is regulated by AKT, and loss of β-catenin activity potentially explains many of the phenotypes described here. The authors show that forced expression of sox10 no longer elevates crestin expression in CAPE-treated embryos and claim that CAPE controls Sox10 activity. However, they provide no evidence for specificity. Would forced expression of another master regulator be similarly disempowered by CAPE expression? The ATAC-seq data is an interesting start, but the authors do not go anywhere with it. By what mechanism does the MITF promoter, but not the PDGFR promoter, become closed upon CAPE treatment? The fact that Sox10 binds both promoters indicates that Sox10 is not likely to be part of the differential effect. What happens to the promoter of Tyrosinase? The authors should probably cite an earlier study showing that 1 μm CAPE, which had no effect on p-AKT, altered the binding of MITF to the Tyrosinase promoter (J Nat Prod. 2013 Aug 23;76(8):1399-405.)

It appears that some markers, e.g., crestin, are globally in NCN while others, e.g. dlx2a and snai1b, are lost just regionally. Is the region where gene expression is lost the same for all markers with regional loss? What is the relationship of time-of-initiation-of-gene-expression and sensitivity to inhibition by CAPE? Is it as simple as CAPE extinguishes all new gene expression? For instance, if CAPE is applied at shield stage, is induction of the neural plate border markers suppressed? If this is the case, then it is misleading to present CAPE as an inhibitor of neural crest induction.

Related to the previous point, a highly relevant study (Pegoraro et al., 2015), oddly not mentioned by the authors but easily discovered through a Pubmed search on "neural crest AKT", it was shown that knockdown of PFKFB4 results in loss of all new gene expression within dorsal embryo, i.e., the domain in which PFKFB4 is expressed, concomitant with loss of p-Akt; all such expression was restored by over-expression of constitutively active Akt. This suggests that the CAPE by reducing p-Akt is a general suppressor of new gene expression, possibly as a consequence of activation (but not completion) of a cell death pathway. The current study offers less mechanistic insight than this previously published work.

It seems plausible that CAPE is simply toxic, by virtue of inhibiting of PI3K and Akt pathways, resulting in reduced gene expression on the way to cell death. In the Pegoraro paper they argue against the possibility by showing that apoptosis could be uncoupled from the defects in gene expression (i.e., by over expression of BCl^-^Xl), but even such an experiment does not rule out the possibility that loss of gene expression results from initiation, but not completion, of an apoptosis program.

[Editors’ note: what now follows is the decision letter after the authors submitted for further consideration.]

Thank you for resubmitting your work entitled "A chemical screen in zebrafish embryonic cells establishes that Akt activation is required for neural crest development" for further consideration at *eLife*. Your revised article has been favorably evaluated by Marianne Bronner (Senior editor), a Reviewing editor, and two reviewers.

All three reviewers are positive about the study and are happy that you have addressed their concerns. Detailed comments from reviewers 1 and 2 are shown below for your information, but after discussion, it was agreed that further experiments are not necessary before publication.

Reviewer #1:

In the revised manuscript Ciarlo and co-authors have addressed, to some degree, my two chief reservations about the first submission. The first was I felt the study did not make a conceptual advance beyond those in the Pegoraro et al.,2015 paper. These authors showed that inhibition of PFKFB4 reduced p-Akt levels and prevented expression of markers of NC and other ectoderm derivatives, and that forced expression of constitutively-active Akt rescued expression of these markers in PFKFB4-morphants. In the revised manuscript Ciarlo et al. now acknowledge the Pegoraro et al. study and point out, legitimately, where their findings differ from it.

The second reservation I had was about the limited mechanistic insight. The revised paper is somewhat improved in this regard because it now presents a clearer model for where CAPE functions. It remains quite mysterious how CAPE affects Sox10 activity at some targets and not at others, but this point is acknowledged in the Discussion it is reasonable to leave this issue for the next paper. The PTEN-overexpression experiment nicely illuminate the aspects of the CAPE-mediated phenotype that are likely to result from inhibition of Akt, which is most but not all of them.

There are additional strengths of the project that will be useful for the field, for instance showing that cultured, GFP-positive NC express appear to express sox10 and foxd3 at about the same level as freshly harvested ones. Another strength is the experimental pipeline of screening compounds that have a greater effect on a tissue specific reporter than ubiquitously expressed one.

Reviewer #2:

The authors have modified the original manuscript adding a few experiments and multiple requested items (better imaging and quantifications) and have improved the manuscript considerably. However, PI3K-AKT was known to modify dorsal ectoderm fates, including NC. The results of this manuscript suggest that AKT effect is restricted to later NC events and does not alter other dorsal ectoderm fates. They propose that animal model differences account for the discrepancy between this zebrafish study and the previous *Xenopus* study. Their data is consistent with this view.

The authors suggest that CAPE acts upstream of PI3K, and confirmed both crestin and pigment cell phenotypes in vivo with various inhibitors. Their postulate that AKT operates differently in these organisms rests on the expression of genes associated with pre- and migratory NC. Yet, they do not provide the effect on these genes with other PI3K and AKT inhibitors. The suggested experiments could confirm their model (probing that AKT does not alter broadly dorsal ectoderm fates. Alternatively, a common effect of AKT in both organisms could emerge, and suggest a distinct AKT-modulation by CAPE.

---

## [Author Response]

[Editors’ note: the author responses to the first round of peer review follow.]

Reviewer #1:This is an interesting study that has identified compounds that interfere with neural crest development using an in vitro-based screening assay followed by validation in vivo in the zebrafish embryo. The study focusses on one compound, CAPE, which shows a dose-dependent reduction in sox10 activity. The authors provide evidence to show that CAPE mediates its effects through inhibition of PI3K/Akt signalling in FGF-stimulated cells, resulting in abnormal neural crest development and some other embryonic defects. The study uses a range of experimental approaches to support the main conclusions. I have a few suggestions for improved clarity or presentation of the data.1) Although the main conclusion is that CAPE acts by reducing Sox10 activity, it should be pointed out more explicitly that the phenotype of CAPE-treated embryos does not resemble that of sox10 mutants closely. The neural crest defects (pigmentation) are less severe, while other aspects (e.g. body curvature and length, pericardial oedema, small head and eyes etc) are more severe. The current text only highlights similarities. The authors point out that CAPE only appears to affect sox10 activity at some promoters, but I think it would be helpful to have show a more detailed comparison with the known effects of the genetic loss of sox10 function. For example, sox10 mutants lack all three pigment cell types, not just melanophores. Are all three pigment cell types reduced in the CAPE-treated embryos, or does treatment only affect the melanophores? (Clearly, some xanthophores are still present, given the yellow colouration of the head in CAPE-treated embryos.) Iridophores should also be checked, since these are missing in sox10 mutants, but are increased at the expense of melanophores in mitfa mutants (Lister et al., 1999).

We agree with the reviewer that a more extensive comparison of CAPE and sox10 mutants is informative. Though CAPE-treated embryos share similarities with sox10 mutants, there are also differences that we have now presented more explicitly in the manuscript. Although many of the early sox10 phenotypes are recapitulated by CAPE, there are more severe neural crest defects in the sox10 mutants, and CAPE has some effects distinct from the sox10 mutants. In sox10 null mutants, all pigment cells are severely reduced. We have now included an analysis of xanthophores and iridophores in CAPE-treated embryos. We found that the xanothophore marker fms was dramatically decreased in CAPE-treated embryos at 24 hpf, but xanothophores recovered by 3 dpf, as indicated by the intense yellow coloring in the head of CAPE-treated embryos (Figure 1, Figure 4—figure supplement 1). Iridophores were decreased at 3 dpf in CAPE-treated embryos, but not as dramatically as melanophores (Figure 4—figure supplement 1). Iridophores also displayed a less dramatic migration phenotype than melanophores based on the fraction of dorsal iridophores. One sox10 mutant, sox10baz1, shows a similar effect in which melanophores are more dramatically reduced than other pigment cells (Delfino-Machin et al., 2017). CAPE likely does not fully inhibit sox10 activity at least at concentrations with no overt toxicity. CAPE leads to some other effects on the body that are not found in sox10 mutants, but overall the early effects of CAPE are rather similar to sox10 mutants. We have adjusted the text to be more explicit about these differences (see Discussion).

2) The other major site of sox10 expression in the embryo is the ear, and sox10 mutant embryos are known to have a very strong ear phenotype (Dutton et al., 2009). Does CAPE treatment also recapitulate these otic defects? If yes, this would support the interpretation that CAPE acts via Sox10, and if no, it might reveal interesting differences in target gene activity for Sox10 between the ear and the neural crest.

We found that CAPE caused an otic vesicle phenotype that was similar to that observed in sox10 mutants. Both caused a subtle change in otic vesicle shape at 24 hpf and a severe defect by 48 hpf including lack of semicircular canal projections and smaller, more closely spaced otoliths. However, we did not observe a change in size of the otic vesicle as in sox10 mutants.

3) With respect to the effects on the embryo such as body curvature, which are unlikely to be attributed to a loss of Sox10 activity, what is the interpretation? Are there any more general effects on cell behaviour? The video showing reduced cell migration is clear, but is there a general inhibition of cell migration? What happens to other migratory cell types e.g. the lateral line primordium?

CAPE indeed has effects on embryonic development that are not related to the neural crest. While it is unclear what target of CAPE is responsible for the body curvature phenotype, it is likely not related to sox10 or Akt activity since embryos deficient in these genes do not display such a phenotype. We found that 5 μM CAPE also slightly inhibited migration of the lateral line primordium based on position of neuromasts at 2 dpf:

This phenotype be related to CAPE’s activity as an Akt inhibitor since Akt is known to play an important role in cellular migration. CAPE likely has an additional non-cell type specific effect on cell migration beyond its effect on sox10 activity/mitfa expression. We have modified the Discussion to include this point.

4) The HOMER analysis (Figure 3) is supposed to identify transcription factors in an unbiased manner, and clearly has generated a lot of data. Only the data for the Sox10 and Mitf motifs are shown, though. It is stated that these are the most enriched, but it would be helpful to show how these compare to say, the next three most enriched in the dataset – otherwise this just looks like confirmation of a result that was already predicted.

We agree with the reviewer that there might have been other sites than sox10 and MITF that would be enriched, but the only sites that were differentially affected were similar to Sox and MITF motifs. We have added the top 10 enriched motifs to the figure, but all of these motifs are similar to Sox or MITF binding sites.

5) It is not clear where the two PI3K inhibitors (described as 'strong hits') appear in Table 1 – please highlight these. PI3K inhibition is not mentioned for any of the compounds in the Target column. More generally, it would help if this Table conveyed more information. For example, do the hits fall into clear groups based on structural similarities?

Table 1 initially indicated only hits that were validated in vivo. We have modified the table to include all hits that validated in vitro and highlighted the hits that validated in vivo.

Reviewer #2:This is a very interesting manuscript in which the authors have identified a specific role for pAKT in the progressive acquisition of NC status. The authors here report another success from an ingenious screen, using a mixture of embryonic cells from early stages of development in which NC have been specified, and remain sensitive to modulation. They identify CAPE as a molecule able to inhibit pAKT when NC development is under way (specified). This inhibition of AKT leads to clear defects in the progression of NC development through alteration of expression of NC genes, reporters, melanocyte formation, and migration. Furthermore, AKT inhibition seems to specifically alter SOX10 activity, which in turn mediates at least some of the other NC effects noted.Their exploration of putative pathway contributions reveals that while CAPE effects on AKT and NC development rely on FGF2 (and perhaps pERK), alternative inhibition of pAKT through other molecules, does lead to AKT inhibition and NC deficits independently of FGF2. This is a solid work with novel insights of signaling that should be very welcomed by the developmental biologists and the neural crest community alike. However a few important concerns are pointed below:The description made for the IH results lacks a) temporal context of expression of specific genes, and b) a proper quantification. The result comments are qualitative and subjective, (for example the authors suggest a robust change on SOX10 via IH, which is and also lack spatiotemporal context. Are early genes spared alterations triggered by CAPE? Does CASPE only alters later NC markers?, all later NC markers? The RNAseq data from sox10:Kaede+ provides quantifiable data, this suggest minimal effect on Sox10, yet significant changes in other genes (Pax3). A better analysis of the regional pattern for IH along with possible limitations of the transgenic line (does it collect all NC?, could subpopulations be missed by this line?) seems warranted, and will be particularly interesting given the ATAC analysis was also performed on this population.

We acknowledge that the gene expression data was presented in a confusing manner and have modified the figure and text to improve it. We have now included the temporal context of gene expression. The genes which had decreased expression were confined to markers of the premigratory and migratory neural crest. Those that are expressed earlier at the neural plate border stage were unchanged. However not all premigratory/migratory neural crest markers were changed, consistent with a block in differentiation of neural crest cells. For example, mitfa and inka1a were decreased, and like crestin these are downstream targets of neural crest specifiers, sox10 and ap2 respectively. The reviewer brings up a good point that using sox10:Kaede+ cells to analyze neural crest gene expression produces inherent biases. The sox10:Kaede transgenic line is more highly expressed in the cranial neural crest, which could contribute to discrepancies in the in situ data compared to the RNA-seq data. Sox10: Kaede is also highly expressed in the otic vesicle, so it is not strictly confined to neural crest. We have now included a statement of these biases in the text.

While describing changes in expression based on IH, the authors bundle SOX10 Crestin and Pax7 as being all dramatically reduced and place other genes like FoxD3, Snai1b, or Inka1a as regionally affected. It is not clear from the images provided that SOX10, or even crestin, are not differentially affected by region. Pax7 instead does seem to be eliminated in normal NC associated regions. In the other hand genes like Pax7, Pax3 and Ets1 seem to increase in Non-NC territories. Additionally, the authors state that genes like Pax3 and Ets1 do not change their profiles in NC territory, yet some of the images provided do not allow a clear conclusion.

We have now included a scoring of the ISH experiments to address the issue of quantification. We found that several of the genes with regional expression changes did not show significant changes and moved them to Figure 2—figure supplement 2 as genes with little to no change upon CAPE treatment. We have also enlarged the panels in Figure 2 to make it easier to see the changes in gene expression upon CAPE treatment. While CAPE did increase the yolk background staining for some probes, this was likely not real expression and did not correlate with whether the gene changed in expression upon CAPE treatment.

It is worth noting that the authors suggest that CAPE and AKT play a role in late events, yet no tests for earlier effects were performed.

We did evaluate the neural plate border genes based on in situ hybridization. Pax3 was unchanged, although there was a change seen by RNA seq. The in situ studies suggest that the neural plate is intact. For this resubmission, we treated embryos with CAPE at shield stage, and studied neural plate border markers by in situ, and there was no major effect on early neural plate development (see comments to reviewer 3).

While addressing cell death and proliferations the authors quantified effects over region when they have the opportunity to specifically address changes over GFP+ neural crest. While the authors suggest that the dramatic effects in cell death arise late, it seems relevant that the cell death increased at 10µM CAPE, the same concentration used through several other experiments.

We agree with the reviewer that cell death is an important consideration for evaluating the mechanism of CAPE since it inhibits Akt signaling. While we did see cell death at 24 hpf with 10 μM CAPE treatment, we used 5 μM CAPE for most experiments beyond 17 ss to reduce developmental defects in the embryos. At this concentration we did not see a substantial increase in cell death up to 48 hpf, but we still saw dramatic changes in mitf and crestin:EGFP expression, indicating that cell death is not responsible for the primary effect of CAPE (Figure 2, Figure 2—figure supplement 3, Figure 3).

The pathway analysis results regarding signaling ligands are not very informative in their current format due to over and under expression of the same pathways. And would suggest to remove this portion.

We have revised the ligand section substantially and hope the reviewer feels that it is adequate explained. Reviewer 3 actually had the opposite request, and asked for more clarifications. We will abide by the editor’s thoughts, but would like to include this information.

The Materials and methods section reports that embryonic cell cultures medium contains 12% FBS and N2 (Insulin 5µg/ml), however they do not provide any arguments or considerations regarding these components while addressing the effect of specific signals at multiple points. Did they remove the N2 while addressing FGF2 or Insulin? Certainly FBS has potential interfering components that should have been noted.

It is indeed surprising that growth factor stimulation has such dramatic effects on cell signaling when a high percentage of FBS is included. However it is perhaps less surprising given that specific growth factors were required for neural crest induction in vitro. We found that FGF and insulin still had major effects on the Akt and Erk pathways on top of the FBS treatment. There are essential components to the FBS necessary for neural crest induction, since it does not occur in the absence of FBS. We have included this in the text. We did remove the N2 for the short time period western experiments in Figure 6 since N2 contains insulin and have included this in the Materials and methods.

It might be relevant to test if other AKT inhibitors (Pi-103 Figure 6—figure supplement 1) could inhibit pAKT triggered by Insulin, FGF2 and iMEK.

We have included additional analysis of the activity of PI-103 under different conditions in Figure 6—figure supplement 1. We found that PI-103 could inhibit p- Akt triggered by Mek inhibition.

The ERK-PI3K analyses were done in zebrafish cell cultures and report broad activity and effects of molecules on signaling. They may not represent signaling effects mediated by NC.

Though the signaling experiments were done on whole embryo cultures, we were able to correlate the signaling changes in whole embryo cultures with crestin:EGFP expression after one day in culture. We cannot be sure that all cells respond in the same manner to growth factors, but based on the dramatic changes in signaling, it is likely the majority of cells.

Regarding the model provided in Figure 6 panel E, while ERK activation might block PIP3, this is context dependent. In the experiments here addressed, it seems that ERK inhibition does boost pPI3K, however under Insulin FGF2 activity, robust pERK does not seem to reduce pAKT, and thus the solid line of inhibition between Erk and PI3P might be misleading.

Based on new data we have revised our model as indicated in Figure 6—figure supplement 1. This new model no longer presents the issue pointed out here. The new model has FGF activating both PI3K/Akt and Mek/Erk, but Mek/Erk inhibits PI3K/Akt, and the net result is no change in p-Akt. This model explains why Mek/Erk stimulation by FGF does not reduce p-Akt.

Reviewer #3:The authors have carried out a chemical screen to identify compounds that reduce expression crestin:gfp in a mixed culture of primary cell derived from dissociated transgenic zebrafish embryos. They focus on one compound, CAPE, and find that it reduces expression of neural crest markers in zebrafish, a few hours before leading to widespread cell death. They provide evidence that CAPE acts upstream of phospho-Akt, a well-known regulator of cell viability.Mechanistic insight from the manuscript is very limited. I would like to see more depth at one level or another. For instance, it is unclear where CAPE acts. The authors mention a paper in which CAPE was shown to inhibit PI3K activity. Is this the authors' preferred model for the principal locus of control of CAPE in the control of gene expression in NC? If so, why does it depend on FGF signaling? The proposed requirement for FGF signaling is not entirely convincing with the data shown. Figure 6 is rather baffling. CAPE is said to be ineffective in the absence of FGF. But even in the absence of added FGF there must be some basal level of RAS/RAF/ERK/MEK signaling, because addition of a MEK inhibitor massively up regulates the amount of phospho-AKT. The bar chart suggests there is some effect of CAPE, but it does not reach significance. Does constitutively active PI3K rescue the effects of CAPE? This seems unlikely because CA-Akt only partially does so, and loss of pten LOF does not seem to do so. [As an aside, the pten mutant data is buried and is under-explained (Figure 6—figure supplement 1). Comparing Figure 6—figure supplement 1,to Figure 3 makes is appear that ptena -/-; ptenb +/- have an elevated level of crestin in comparison to wild-types. Is that true? Do the double mutants have other phenotypes accompanying the elevated crestin?] Returning to the question of the locus of CAPE activity, does CAPE work to reduce p-AKt levels in ptena/b double mutants? Do the PI3K inhibitors work to rescue elevate crestin in such double mutants? If not, then perhaps PI3K really is the central locus of control. There is more to learn from this mutant that would help point to the locus of CAPE action.

This paragraph deals with a number of issues, and we thank the reviewer for pointing these out.

1) We have modified our model of how CAPE acts to inhibit Akt signaling. CAPE likely acts downstream of FGF stimulation as indicated in Figure 6—figure supplement 1. This model is consistent with all of our data, including the new data in Figure 6—figure supplement 1.

2) Regarding the pten mutant data, when we added the second replicate of this experiment to increase the n number (Figure 6—figure supplement 1) we saw that there is no significant difference in crestin expression between the wild type and pten mutant embryos.

Another option for mechanistic insight that would elevate the significance of the study would be to dig deeper into the transcriptional regulation. What causes the chromatin to close at the MITF promoter? As the authors noted, Β-catenin is regulated by AKT, and loss of β-catenin activity potentially explains many of the phenotypes described here. The authors show that forced expression of sox10 no longer elevates crestin expression in CAPE-treated embryos and claim that CAPE controls Sox10 activity. However, they provide no evidence for specificity. Would forced expression of another master regulator be similarly disempowered by CAPE expression? The ATAC-seq data is an interesting start, but the authors do not go anywhere with it. By what mechanism does the MITF promoter, but not the PDGFR promoter, become closed upon CAPE treatment? The fact that Sox10 binds both promoters indicates that Sox10 is not likely to be part of the differential effect. What happens to the promoter of Tyrosinase? The authors should probably cite an earlier study showing that 1 μm CAPE, which had no effect on p-AKT, altered the binding of MITF to the Tyrosinase promoter (J Nat Prod. 2013 Aug 23;76(8):1399-405.)

Though the ATAC-seq data was useful in providing a clue to relevant transcription factors for CAPE’s activity, changes in chromatin accessibility were likely not the primary mechanism by which CAPE works. We have modified Figure 4 to more clearly indicate this. We show that there was no change in ATAC-seq signal at the crestin promoter, though crestin expression was clearly decreased. We also did not see a reproducible change on the tyr promoter:

Chromatin closing at the mitfa promoter could be secondary to other transcriptional effects of CAPE.

We have added a reference in the Discussion to the J. Nat. Prod. paper mentioned.

We did not find any difference in β catenin phosphorylation (S552) or level with CAPE treatment, so that likely does not explain the effects of CAPE:

We agree with the reviewer that another transcription factor was needed to show specificity for CAPE affecting sox10 activity. We have added the effect of tfap2c RNA injection on crestin:EGFP expression (Figure 3—figure supplement 1). Though tfap2c does not have as dramatic of an effect on crestin:EGFP expression as sox10, we found that it could increase crestin:EGFP expression in CAPE-treated embryos, unlike sox10.

It appears that some markers, e.g., crestin, are globally in NCN while others, e.g. dlx2a and snai1b, are lost just regionally. Is the region where gene expression is lost the same for all markers with regional loss? What is the relationship of time-of-initiation-of-gene-expression and sensitivity to inhibition by CAPE? Is it as simple as CAPE extinguishes all new gene expression? For instance, if CAPE is applied at shield stage, is induction of the neural plate border markers suppressed? If this is the case, then it is misleading to present CAPE as an inhibitor of neural crest induction.

To confirm that CAPE does not simply prevent new gene expression, we did an additional experiment. We treated embryos with CAPE at shield stage and evaluated neural plate border marker expression at 2 ss (Author response image 4). We found that while 10 μM CAPE was toxic to embryos at this stage, 5 μM CAPE did not change expression levels of pax3a and dlx3b and only slightly decreased anterior levels of msx1b. In contrast, we saw a dramatic reduction in crestin:EGFP, mitfa, and pigment cell gene expression with 5 μM CAPE treatment. Therefore it is unlikely that CAPE simply prevents new gene expression. We have adjusted the main Results section that CAPE’s effect on gene expression was restricted to markers of premigratory/migratory neural crest and not those that are expressed earlier at the neural plate border, at least of the genes we tested.

Related to the previous point, a highly relevant study (Pegoraro et al., 2015), oddly not mentioned by the authors but easily discovered through a Pubmed search on "neural crest AKT", it was shown that knockdown of PFKFB4 results in loss of all new gene expression within dorsal embryo, i.e., the domain in which PFKFB4 is expressed, concomitant with loss of p-Akt; all such expression was restored by over-expression of constitutively active Akt. This suggests that the CAPE by reducing p-Akt is a general suppressor of new gene expression, possibly as a consequence of activation (but not completion) of a cell death pathway. The current study offers less mechanistic insight than this previously published work.

We agree with the reviewer that this article is highly relevant to our study and have added a discussion of it to the manuscript. The effect of CAPE that we observed was distinct from that observed by Pegoraro et al. with PFKFB4 loss. Pegoraro focused on the effect of Akt signaling on the early specification of dorsal ectoderm derivatives, while we saw that CAPE affected differentiation of neural crest derivatives. We observed effects on genes downstream of sox10 such as crestin and mitfa, but little to no change in most premigratory neural crest markers (snai1b, foxd3, pax3a, tfap2a, tfap2c) or the neural marker *Sox2*. This would most certainly not be the case if CAPE prevented specification of dorsal ectoderm derivatives. Normal neural crest specification was also indicated by the indistinguishable pattern and levels of sox10:GFP in control and CAPE-treated embryos prior to neural crest migration. Pegoraro et al., on the other hand, saw a failure of neural crest specification as indicated by absence of sox10 and snai2.

Both our study and that of Pegoraro show that Akt promotes cellular differentiation independently of cell death but in different developmental contexts. A comparison of the two studies suggests that there are species-specific differences in the role of Akt in ectoderm development between zebrafish and frogs. We have added these points to the Discussion. We have also changed the wording of the manuscript to indicate that the novel role of Akt is in neural crest differentiation.

As mentioned previously, CAPE did not alter expression of neural plate border markers even when embryos were treated at shield stage.

It seems plausible that CAPE is simply toxic, by virtue of inhibiting of PI3K and Akt pathways, resulting in reduced gene expression on the way to cell death. In the Pegoraro paper they argue against the possibility by showing that apoptosis could be uncoupled from the defects in gene expression (i.e., by over expression of BCl^-^Xl), but even such an experiment does not rule out the possibility that loss of gene expression results from initiation, but not completion, of an apoptosis program.

While it is certainly a concern that inhibition of Akt could lead to non-specific effects related to cell death, we note that 5 μM CAPE still dramatically reduced crestin:EGFP, mitfa, and pigment cell gene expression while having little effect on cell death up to 48 hpf. Therefore, early effects of cell death are likely not responsible for the effect of CAPE on neural crest gene expression.